# Welfare-Optimal Classification with Accuracy Auctions

**Bana Sadi** [* 1]  **Eden Saig** [* 1 2]  **Nir Rosenfeld** [1]

## Abstract

Prediction algorithms are increasingly used to inform decisions about humans, but maximizing accuracy—the standard learning objective—does not necessarily maximize user benefits. Instead, we propose optimizing social welfare, defined as the average gain users receive from correct predictions. Welfare enables to express, and therefore account for, heterogeneity in how much users benefit from accuracy. But since these valuations are private and users can gain from overreporting them, learning must simultaneously elicit truthful values and optimize welfare with respect to them. To this end, we propose a novel learning algorithm that incorporates a truthful auction. We show how to compute allocations and prices efficiently, and bound the number of paying users—which surprisingly is independent of the sample size. We conclude with experiments on real and synthetic data that demonstrate our algorithm and explore the connections between welfare and accuracy.

## 1. Introduction

The social domain has much to gain from the broad capacity of machine learning to generate accurate predictions effectively and at scale. Often, the direct beneficiaries of accuracy are the individuals who are the object of prediction; consider diverse examples such as online content recommendation, career planning advice, asset price assessment, educational program assignment, and medical condition diagnosis. This makes the goal of maximizing expected accuracy—the standard objective of supervised learning—a sensible target. But while maximizing expected accuracy serves to increase the number of correct predictions, it remains silent about *which*

individuals will receive them—and which will not. In this sense, the learning objective is underspecified with regards to the distribution of errors across users in the population.

How then should we determine where to focus accuracy efforts? One answer, promoted by the literature on fairness in learning, is to strive for equity, for example across social groups (Dwork et al., 2012) or between individuals with similar traits (Zemel et al., 2013). While suitable for some settings, one drawback is that imposing equity (e.g., via fairness constraints) often reduces overall predictive performance (Kleinberg et al., 2016; Menon & Williamson, 2018), and consequently, the social benefit of learning. Another is that by focusing on equity, fairness fails to capture the idea that individuals can differ in how much they stand to gain from correct predictions, or lose from incorrect ones.

As an alternative, this work advocates designing learning algorithms that maximize the expected benefit of users from accurate predictions. Adopting an economic perspective, and echoing similar ideas expressed recently (Heidari et al., 2018; Hu & Chen, 2020; Rosenfeld & Xu, 2025; Jordan, 2025), we treat accuracy as an explicit *limited resource* that necessitates deliberate allocation. Our key modeling assumption is that users can differ in how much value they derive from correct predictions. Given these individual values, the learning objective is to find a classifier that maximizes *expected social welfare*, defined as the population-average benefit from accurate predictions. This formalizes accuracy allocation as a realizable, well-defined objective for maximizing social benefit through learning. We refer to this new task as *welfare-optimal classification*.

The main conceptual challenge in maximizing welfare through classification is that values are inherently private to users, and so cannot be observed by the system. Since accuracy is a contested resource, users are prone to overreporting their values, in hopes that this steers learning outcomes in their favor. Thus, a system interested in maximizing welfare cannot simply treat values as additional observed variables (i.e., as "just another feature"). Rather, it must contend with this informational gap by simultaneously eliciting users' true values, and optimizing for expected social welfare under them. Neglecting to do so, e.g., by using conventional learning approaches, amounts to assuming implicitly that values are uniform, which leads to suboptimal allocations.

---

[*]Equal contribution, alphabetical order.  [1]Department of Computer Science, Technion, Haifa, Israel [2]Department of Computing and Mathematical Sciences, Caltech, Pasadena, USA. Correspondence to: Bana Sadi <sadi.bana@campus.technion.ac.il>, Eden Saig <edens@caltech.edu>, Nir Rosenfeld <nirr@technion.ac.il>.

*Proceedings of the 43[rd] International Conference on Machine Learning*, Seoul, South Korea. PMLR 306, 2026. Copyright 2026 by the author(s).

Our main contribution is a learning approach which tackles the above dual challenges. The core idea is to design a learning objective that also serves as a mechanism for incentivizing truthful reporting. Our approach works in two stages: first running an auction for eliciting values, and then training a classifier to maximize the corresponding social welfare objective. Since the items allocated are correct predictions, we refer to the former as an 'accuracy auction'. The two stages are linked in that the auction's objective derives from the learning task, and the set of feasible allocations are restricted to those expressible by classifiers in the chosen model class. This presents challenges—but also provides structure, which we exploit for devising efficient schemes for computing allocations and payments.

The information gap due to private user information means that optimizing welfare entails a cost necessary for aligning incentives. In auctions, this takes the form of individual payments collected from (some) users. Our approach restricts the extent and potential burden of payments in two ways. First, payments are only required at train time; once a suitable classifier has been found, generalization ensures that the attained level of social welfare will hold indefinitely and for free for all test-time users. Thus, there is no need for (costly) elicitation after deployment, and maximizing welfare admits a fixed and bounded total cost.

Second, our main theoretical result states that for many common linear classification-stable learning algorithms, the number of paying users is bounded by a constant, and so does not grow with the number of training samples This constitutes a large class of learning algorithms, including SVM and logistic regression. The proof builds on a novel connection between auctions, learning, and algorithmic stability, shedding light on the mechanism underlying this phenomenon and the forces at play. We then complement this result by establishing a lower bound showing that for the $k$-NN algorithm, the number of paying users is at least linear in the number of samples when the data distribution contains regions with label noise. Together, these suggest that minimizing payments is possible, but requires careful consideration of the choice of learning algorithm.

We conclude with experiments that shed light on how accuracy auctions affects prices, welfare, and accuracy, and how these notions relate. Our results support our theory, showing that the sum of payments and the number of paying users is very small in practice—often just a handful, even for large high-dimensional data. This bears positive social implications, but also poses questions as to which users will be subject to this burden. We also study how accuracy and welfare trade off, using real data and leveraging labels and features to simulate personalized values. We demonstrate how learning has the capacity to improve welfare, often with little reduction in accuracy, by investing in eliciting true user values.

## 2. Related Work

Our work relates to the growing field of strategic learning, wherein learning must contend with strategic user behavior. A canonical task in this field is *strategic classification* (Hardt et al., 2016; Levanon & Rosenfeld, 2021), in which learning aims to maximize accuracy, but users can manipulate their features, at a cost, to obtain favorable predictions. Here it is assumed that feature information is private to users—but only at test time, and that a cost function, rather than a mechanism, limits the degree of misreporting. Interestingly, untruthful behavior can in some cases improve accuracy, due to a causal relations (Miller et al., 2020; Bechavod et al., 2022; Horowitz & Rosenfeld, 2023), social network structure (Eilat et al., 2023), externalities and market forces (Sommer et al., 2025; Hossain et al., 2025), or when user and system incentives align (Levanon & Rosenfeld, 2022).

Earlier formulations of strategic learning consider labels as private (Meir et al., 2012), whereas others allow agents to withhold certain features (Krishnaswamy et al., 2021), attributes (Nair et al., 2022), or their participation altogether (Horowitz et al., 2024). Lacking an exogenous force limiting misreporting, these settings are similar to ours in that they require a mechanism that incentives truthful behavior. However, none of these consider user values as private, nor allow for variability in valuations. In Sundaram et al. (2021), users can have individual values, but these are observed at train time together with features and labels. Furthermore, the objective—as in most works—is accuracy, not welfare.

While algorithmic fairness has been popular as a means to reason about social outcomes in learning, recent voices have proposed social welfare as a viable alternative framework (Heidari et al., 2018; Hu & Chen, 2020; Rambachan et al., 2020; Kasy & Abebe, 2021; Cousins, 2023; Jordan, 2025). Though fairness sometimes views accuracy as a limited resource (Gölz et al., 2019), it controls allocation indirectly through hard constraints, typically based on group membership or similarity measures. In social welfare, resources are made explicit, and predictions inform allocations directly and following a social planner's policy (Rolf et al., 2020; Shirali et al., 2024; Fischer-Abaigar et al., 2025). Additionally, the economic foundations of welfare lend naturally to modeling and accounting for agency and strategic behavior within the learning objective (Rosenfeld & Xu, 2025). In our setting, once incentives have been aligned and valuations elicited, the task reduces to example-dependent cost-sensitive learning (Elkan, 2001; Zadrozny et al., 2003).

From a game-theoretic perspective, our work extends the theory of single-parameter auction design (Myerson, 1981) to settings in which allocations are induced by learning algorithms. Our main contribution in this context is to show that a broad family of learning algorithms induces truthful auctions via monotonicity, and analyze their welfare properties.

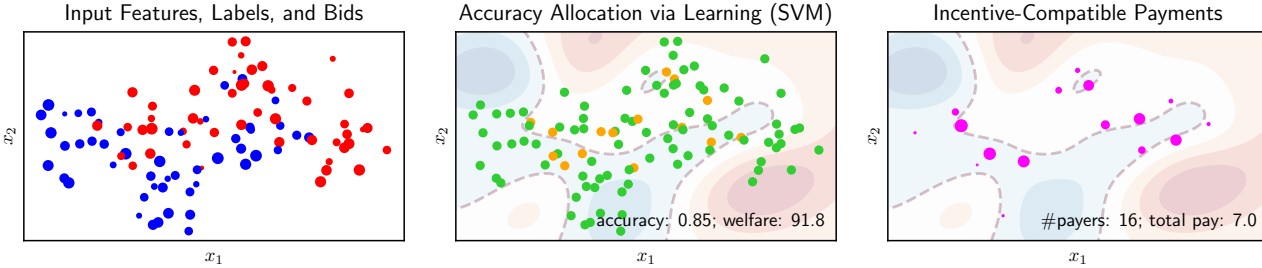

*Figure 1.* Accuracy auction induced by a classification algorithm. **(Left)** Each training point represents a participant in the auction: color represents label, and size represents value of accurate prediction. **(Center)** Welfare-maximizing allocation of pointwise accuracy via weighted loss minimization, showing learned score function (background color), decision boundary (dashed line), and prediction correctness (point color). **(Right)** User payments (size depicts magnitude) guarantee truthful elicitation, despite being highly sparse.

## 3. Setup

Users in our setting are represented by features $x \in \mathcal{X} \subseteq \mathbb{R}^d$, labels $y \in \{\pm 1\}$, and values $v \in \mathbb{R}_+$, and the user population is given by an unknown joint distribution $D$ over triples $(x, y, v)$. We assume users know their features $x$ and values $v$, but not their label $y$, and look to the system to provide them with accurate predictions $\hat{y} \approx y$. The gain of users from accurate predictions is given by their *utility function*:

$$u(y, \hat{y}; v) = v \cdot \mathbb{1}\{y = \hat{y}\} \tag{1}$$

Thus, all users gain from correct predictions, but can differ in how much this benefits them. For clarity, Eq. (1) assumes erroneous predictions always entail zero value; in Appendix A.2 we show this is made w.l.o.g., and discuss how our results naturally extend to support individualized values for both correct and incorrect predictions. We also assume that $v \in [0, V]$ for some maximal value $V \in \mathbb{R}_{\geq 0}$.

The goal of learning in our setting is to maximize the benefit of prediction to the population of users. This is formulated via the objective of maximizing *expected social welfare*:

$$W(h) = \mathbb{E}_D[u(y, h(x); v)] \tag{2}$$

i.e., the utility users gain from predictions $\hat{y} = h(x)$, by selecting a classifier $h$ from a chosen model class $H$. We focus on classes of score-based classifiers in the form of $h_f(x) = \text{sign}(f(x))$ where $f : \mathcal{X} \to \mathbb{R}$ is a parametric score function. To optimize Eq. (2), and as is common in learning, we assume the system can collect a labeled sample set $S = \{(x_i, y_i)\}_{i=1}^m \sim D_{XY}^m$ to use for training $h$.

Importantly, the system does *not* have sample access to values $v_i$, as these are private to users—who can chose how to report (or misreport) them at their own discretion. The challenge in learning is therefore to maximize welfare despite this information gap, in addition to the standard computational and statistical challenges that Eq. (2) poses.

**Eliciting user values.** A natural approach to optimizing Eq. (2) is to first obtain the true values $\boldsymbol{v} = (v_1, \ldots, v_m)$ of

users in the training set $S$. This can be achieved by employing an appropriate mechanism—in our case, an auction—that incentivizes truthful reporting. Auctions help allocate limited resources by first collecting bids $b_1, \ldots, b_n$ from all participants, and then based on these bids, determining who gets which resources and at what price. These notions are defined formally in Sec. 4.1. We will assume users are rational, i.e., bid to maximize their expected payoff (namely utility minus cost) from allocations, which may have randomness. An auction is *truthful* if it incentivizes users to bid their true values, $b_i = v_i$, i.e., it is in the best interest of each user to report truthfully. We focus on truthful auctions whose objective is to allocate resources—namely accurate predictions—in a way which maximizes social welfare.

**Welfare vs. accuracy.** For the special case of uniform valuations, $v = 1$, Eq. (2) matches the conventional learning objective of maximizing accuracy. This implies that applying standard techniques for learning a classifier amounts to assuming (perhaps implicitly) that all users benefit equally from correct predictions. While capturing a certain notion of equity, a simple construction shows that maximizing accuracy can result in arbitrarily bad welfare (see Appendix D.1). On the other hand, the naïve approach of simply asking users to reveal their private values will result in all users reporting $v = V$, the maximal value, as this best advances their own self-interest; this result follows from our claim in Thm. 1. Thus, neglecting to account for user incentives in learning leads to the same outcomes as working with uniform values, which further motivates the need for a mechanism.

**The cost of welfare.** Aligning incentives via mechanism design typically requires enabling monetary transfers. In auctions, these manifest as user payments $p_i \in \mathbb{R}$. We aim for auctions that satisfy *individual rationality*, i.e., in which payments never exceed values. This ensures that all users with strictly positive values are incentivized to participate in the auction. Payments in our setting can either be charged directly, or implemented to indirectly reduce utility (e.g.,

by injecting noise). Since our goal is to maximize social welfare, payments can be viewed as a deadweight loss that is undesired yet necessary for incentive compatibility; for a discussion on the necessity of such "money burning", see Hartline & Roughgarden (2008). Nevertheless, our results show that common learning algorithms induce accuracy auctions achieving near budget-balance, and in which the gap between residual and social surplus is often negligible. Furthermore, and as noted, even this small gap manifests only at train time, while test-time users bear no costs indefinitely.

Our setting is illustrated in Figure 1.

# 4. Accuracy Auctions

Our general approach to maximizing expected welfare will be to optimize an empirical proxy objective based on values $v_i$ elicited through a truthful auction. From Eq. (1), and given data $S$, we can rewrite the empirical analog of Eq. (2) as an equivalent weighted loss minimization objective:

$$\operatorname*{argmin}_{h \in H} \frac{1}{m} \sum_{i=1}^{m} v_i \mathbb{1}\{y_i \neq h(x_i)\} \qquad (3)$$

Since minimizing the 0-1 loss is computationally challenging, and as is common in learning, the next step is to replace it with a tractable surrogate $\ell$ (e.g., hinge loss). Adding a regularization term $R$ (e.g., $L_2$) to control overfitting gives:

$$\operatorname*{argmin}_{h \in H} \frac{1}{m} \sum_{i=1}^{m} v_i \ell(y_i, f(x_i)) + \lambda R(f) \qquad (4)$$

This will be our target objective. We use $\mathcal{O}(h; \boldsymbol{w})$ to denote the above objective with general example weights $\boldsymbol{w} = (w_1, \ldots, w_m)$. Interestingly, we will see that different loss functions have different implications for auctions.

If we are able to gain access to $\boldsymbol{v}$ and learn a classifier $h$ that both minimizes the loss in $\mathcal{O}(h; \boldsymbol{v})$ and generalizes well, then $h$ will gain high expected social welfare on the entire population. We therefore now turn to the question of how to elicit true values as a first step towards optimizing Eq. (4).

## 4.1. Auctions

Auctions are mechanisms for effectively allocating limited resources to users on the basis of their submitted bids. Formally, an auction is a defined by a pair of functions $(a, p)$ that map a vector of user bids $\boldsymbol{b} = (b_1, \ldots, b_m) \in \mathbb{R}^m$ to corresponding *allocations* $a(\boldsymbol{b}) = \boldsymbol{a} = (a_1, \ldots, a_m) \in \mathcal{A} \subseteq \{0, 1\}^m$, where $\mathcal{A}$ is a set of feasible allocations, and individual *payments* $p(\boldsymbol{b}) = \boldsymbol{p} = (p_1, \ldots, p_m) \in \mathbb{R}^m$. An auction proceeds in three steps: (1) the auctioneer commits to and publishes the pair $(a, p)$, (2) users submit their bids $b_1, \ldots, b_m$, and (3) the auctioneer computes $a(\boldsymbol{b})$ and $p(\boldsymbol{b})$, distributes allocations, and collects payments. For a given $(a, p)$, we use $\boldsymbol{b}_{-i}$ to denote all bids in $\boldsymbol{b}$ except $i$'s, and $a_i(b_i; \boldsymbol{b}_{-i})$ and $p_i(b_i; \boldsymbol{b}_{-i})$ as user $i$'s allocation and payment, respectively, as a function of her own bid $b_i$ for fixed $\boldsymbol{b}_{-i}$.

**Allocation by classification.** Users in our setting are interested in accurate predictions (Eq. (1)), but typically no classifier exists that can predict perfectly for everyone. This makes accuracy itself a (homogeneous) limited resource which inevitably requires allocation. In our setting, allocations are induced by the choice of classifier; with slight abuse of notation we write $a_i(h) = \mathbb{1}\{y_i = h(x_i)\}$ and $\boldsymbol{a}(h) = (a_1(h), \ldots, a_m(h))$. Note this implies that not all allocations are feasible, since correct predictions are determined by the choice of classifier $h$. Hence, the set of feasible allocations $\mathcal{A} = \mathcal{A}(H) \subseteq \{0, 1\}^m$ is induced by the model class $H$. This suggests that the expressivity of $H$ affects the flexibility of resource allocation.

## 4.2. Allocating Accuracy

Recall our goal is to maximize welfare by allocating accuracy. By definition, allocations $\boldsymbol{a}^*$ derived from the optimal solution $h^*$ to Eq. (3) achieve this. We aim to approximate $\boldsymbol{a}^*$ by solving $\mathcal{O}(h; \boldsymbol{w})$ with $\boldsymbol{w} = \boldsymbol{v}$ attained via auction. Given the above, it is natural to define the auction's allocation rule as:

$$a(\boldsymbol{b}) = \boldsymbol{a}(\hat{h}), \qquad \hat{h} = \operatorname*{argmin}_{h \in H} \mathcal{O}(h; \boldsymbol{b}) \qquad (5)$$

i.e., as the accurate predictions obtained from training with example weights $\boldsymbol{b}$. Thus, if users bid truthfully, then the auction (approximately) optimizes social welfare.

**Incentive compatibility.** Our main result here is that the above $a(\boldsymbol{b})$, coupled with an appropriate payment rule $p(\boldsymbol{b})$, can incentivize truthful reporting. The key lies in proving that such allocations are *monotone* in individual user bids. This then allows us to invoke Myerson's Lemma (Myerson, 1981) to obtain guarantees for truthfulness, as well as a template for an appropriate payment scheme.

**Theorem 1** (Monotonicity). *Let $a(\boldsymbol{b})$ be as in Eq. (5). Then for all $i \in [m]$ and any fixed $\boldsymbol{b}_{-i}$, the allocation $a_i(b_i; \boldsymbol{b}_{-i})$ for user $i$ is (weakly) increasing in her bid $b_i$.*

Proof in Appendix A.5. Intuitively, the result states that increasing an example's weight in the (proxy) objective can only improve its (true) accuracy from the resulting learned classifier. The result applies to any score-based model class and to all margin-based losses (e.g., hinge loss, log-loss, 0-1 loss). The technical challenge lies in showing that monotonicity is preserved when weights are pushed through the argmin operator applied to $\mathcal{O}(h; \boldsymbol{b})$. Interestingly, weak local conditions on the proxy loss suffice for each loss component to become concave and maintain monotonicity. Leveraging the monotonicity guarantees of Theorem 1, we invoke Myerson's lemma to obtain the auction mechanism:

**Corollary 1.** *There exists a payment rule $p(\boldsymbol{b})$ such that the auction $(a, p)$ is dominant-strategy incentive-compatible (DSIC), i.e., each user's optimal strategy is to bid truthfully, $b_i = v_i$, irrespective of all other users' strategies.*

**Payments.** Myesron's lemma also gives an explicit formula for an appropriate (and unique) payment rule:

$$p_i(b_i; \boldsymbol{b}_{-i}) = \int_0^{b_i} z \frac{\mathrm{d}}{\mathrm{d}z} a_i(z; \boldsymbol{b}_{-i}) \, \mathrm{d}z \qquad (6)$$

Since allocations in our case are binary, and due to monotonicity, the above can be equivalently expressed as the solution to a constrained optimization problem:

$$p_i(b_i; \boldsymbol{b}_{-i}) = \mathrm{argmin}_{z \in [0, b_i]} \ z \ \text{ s.t. } \ a_i(z; \boldsymbol{b}_{-i}) = 1 \quad (7)$$

i.e., the minimal bid $z$ user $i$ requires to ensure a correct prediction. If no such $z$ exists, then we define $p_i = 0$. Given this interpretation, we refer to the solution of Eq. (7) as the *critical bid* for user $i$, denoted $b_i^c := p_i(b_i; \boldsymbol{b}_{-i})$.

**Individual rationality.** For train time users, payoffs are given by their utility (Eq. (1)) minus payments (Eq. (6)). Note Eq. (6) implies wrong predictions entail $p_i = 0$. If we further assume $b_i = 0$ implies $p_i(\boldsymbol{b}) = 0$, then Myerson's Lemma guarantees *individual rationality* (IR), i.e., truthful agents will never have negative payoffs since $p_i \leq v_i$.

## 5. Payment Analysis

We now turn to analyzing the surplus properties of accuracy auctions. Since truthful reporting implies approximate welfare maximization, our key object of interest will be the expected sum of payments extracted from users, defined as:

$$\mathrm{pay}_m(\mathcal{A}) = \mathbb{E}_{S \sim D^m} \left[ \sum_{i=1}^m p_i(\boldsymbol{v}; \mathcal{A}) \right] \qquad (8)$$

Our goal is to understand how total payments scale with $m$, and how this burden is distributed across users.

Individual rationality implies that users never pay more than their valuations, and thus $\mathrm{pay}_m(\mathcal{A}) \leq \mathbb{E}[\sum_i v_i] \leq mV$. From this, we might expect the sum of the payments to grow with $m$ in general. However, and perhaps surprisingly, here we prove that for $L_2$-regularized risk minimization over linear classifiers, payments are upper-bounded by a constant, $\mathrm{pay}_m(\mathcal{A}_{\mathrm{lin}}) = O(1)$, and so are independent of $m$. We then complement this with a lower-bound showing that payments due to $k$-Nearest-Neighbor ($k$-NN) classifiers exhibit at least linear growth, $\mathrm{pay}_m(\mathcal{A}_{k\text{-NN}}) = \Omega(m)$ when label noise is present in some region. Together, these establish the largest possible asymptotic discrepancy in payments.

### 5.1. Payments Upper Bound for Linear Classifiers

We begin by showing that the expected sum of payments of accuracy auctions using linear classifiers is bounded by a constant. Our results apply to regularized risk minimization (Eq. (4)) with $L_2$ regularization and any proxy loss that is convex, $\sigma$-Lipchitz in $f(x)$, and differentiable near the origin, covering most common loss functions. We make two regularity assumptions on the data distribution: that features are bounded in norm and probability density, and that the distribution has non-trivial signal strength, meaning that the theoretical optimal classifier is non-zero.

**Theorem 2** (Linear RRM payment upper bound; Informal). *The expected sum of payments of any accuracy auction induced by regularized risk minimization over a linear class is upper-bounded by a constant for any sample size $m$:*

$$\mathrm{pay}_m(\mathcal{A}_{\mathrm{lin}}) = O(1)$$

Formal statement and proof in Appendix A.6. The main tool for establishing our bound is *algorithmic stability* and its application to supervised learning (Bousquet & Elisseeff, 2002). We outline the key components of the proof below.

**Stability.** Broadly, a learning algorithm is stable if the effect of modifying a single point from the sample set on learning outcomes (e.g., loss or example scores) diminishes with the size of the sample set. The common use of stability in learning is generalization; here we use it for bounding payments. For our proof, we use the notion of *classification stability*, which bounds the difference in example scores:

**Definition 1** (Classification stability; Bousquet & Elisseeff (2002)). Let $H$ be a collection of classifiers $h_f(x)$ induced by score functions $\mathcal{F} \subseteq \mathbb{R}^{\mathcal{X}}$. Let $\mathcal{A}$ be a learning algorithm mapping sample sets $S$ of size $m$ to score functions $f \in \mathcal{F}$, denoted $f_S = \mathcal{A}(S)$. Then $\mathcal{A}$ is $\beta$-*classification stable* if:

$$\forall S \quad \forall i, j \in [m] \quad |f_S(x_j) - f_{S^{\setminus i}}(x_j)| \leq \beta \qquad (9)$$

where $S^{\setminus i}$ denotes the sample set without the $i^{\text{th}}$ example.

In their canonical paper, Bousquet & Elisseeff (2002) show that the SVM algorithm for example has classification stability $\beta_{\mathrm{SVM}} \leq \kappa^2 / 2\lambda m$, where $\kappa$ bounds the kernel diameter as $\sqrt{K(x, x)} \leq \kappa$, and equivalently $\max_i \|x_i\|_2 \leq \kappa$ for linear kernels when features are bounded. For weighted objectives, this notion of stability naturally extends by associating $S^{\setminus i}$ with a modified dataset that sets a zero weight to the $i^{\text{th}}$ example $w_i \leftarrow 0$, and standard stability is a special case of weighted stability with weights $w_i = 1$. Moreover, we prove that the classification stability bound of SVMs also extends naturally when weights are bounded:

**Lemma 1.** *The SVM classification algorithm with weighted objective and bounded weights $w \in [0, V]$ has classification stability bounded by $\beta \leq V\kappa^2 / \lambda m$.*

Proof in Appendix A.6.3, following a technique similar to the unweighted result. An important implication of Lemma 1 is that weighted objectives maintain the same

stability rates, i.e., weighted SVM still has $\beta = O(1/m)$. These results naturally extend to $\sigma$-Lipschitz loss, entailing a factor of $\sigma$. Interestingly, the weighted objective allows for tighter *individual* stability bounds, where the effect of removing an example $i$ on all others is at most $\beta_i \leq v_i \sigma \kappa^2 / \lambda m$. This will later prove useful for computational speed up in Sec. 6.

**Necessary condition for payment.** Stability entails a particular structure on which type of users can possibly pay. By Eq. (7), and from the monotonicity of $a$ (Thm. 1), a user $i$ will have $p_i > 0$ only if $a_i(b_i; \boldsymbol{b}_{-i}) = 1$ and $a_i(0; \boldsymbol{b}_{-i}) = 0$, i.e., if $i$ gets a correct prediction under the default objective $\mathcal{O}(h; b_i, \boldsymbol{b}_{-i})$, but removing the example results in a learned classifier that errs on $x_i$. By definition, this is possible only if $y_i f(x_i)$ changes from positive to negative when $w_i$ is changed from $b_i$ to 0. Stability then implies that this is possible only for points with sufficiently low scores.

**Lemma 2.** *A point $x_i$ might pay only if $y_i f(x_i) \in (0, \beta_i]$.*

This result will also be key for our algorithm in Sec. 6.

**Total payments.** To bound the sum of payments, the final step is to take an expectation over sample sets $S \sim D^m$. The proof relies on the stability bounds presented above, together with notion of non-trivial signal strength, and superlinear tail bounds for the learned feature vector norm. We begin by applying Lemmas 1 and 2 to decouple the training set from the point under consideration, and bound the probability of paying by a constant which is proportional to the boundedness of features, and inversely proportional to both $m$ and the norm of the learned feature vector. We then combine the non-trivial signal assumption with boundedness to obtain an exponential tail bound on the norm of the learned feature vector, yielding a constant payment bound.

**Geometric interpretation.** By Lemma 2, a user will possibly pay only if $x_i$ lies on the correct side and sufficiently close to the decision boundary. Note this condition is necessary, but not sufficient, and generally we expect that many (if not most) of these users will not need to pay. Geometrically, payments can therefore be incurred only in a band of width $\beta$ near the decision boundary. Increasing the number of samples $m$ entails two opposing effects: On the one hand, the number of points that fall within the band can grow linearly. On the other hand, stability implies that the band shrinks at rate $1/m$. Thus, the effects cancel out.

**Economic implications.** Although Thm. 2 states that the number of paying users is at most constant, this does not mean that payments are distributed uniformly across the population. Rather, Lemma 2 implies that payments concentrate on marginal users—typically those that are 'hard' to classify. Thus, susceptibility to payment is not a property of an individual's features, labels, or value measured in isolation, but rather depends on their position relative to the classifier and data distribution. Payments can be viewed of as a means for users with incorrect predictions to influence the classifier into flipping their predictions to be correct. A user will pay only if doing so is both necessary to secure a correct prediction—and possible. In this sense, higher values give users greater leverage over outcomes.

### 5.2. Payments Lower Bound for $k$-NN

For the converse result, we analyze the asymptotic behavior of the $k$-Nearest-Neightbor algorithm, showing a linear lower bound on expected payments. Here we make the mild assumption that there exists some bounded region within the feature space in which some amount label noise exists.

**Theorem 3** ($k$-NN payments lower bound; Informal)**.** *Consider a data distribution $D$ with some amount of label noise in some region. Then for a sufficiently large constant $k$, the expected sum of payments of an accuracy auction induced by a $k$-NN classifier grows linearly with the sample size $m$:*

$$\mathrm{pay}_m(\mathcal{A}_{k\text{-NN}}) = \Omega(m)$$

Formal statement and proof in Appendix A.7. The proof leverages a classic result from the theory of *probabilistic optimal partitioning* (Karmarkar et al., 1986). Intuitively, consider a set of $k$ scalars $w_1, \ldots, w_k$ sampled from a sufficiently-smooth distribution. The surprising result of Karmarkar et al. (1986) shows that with probability at least $1/2$ over the random draw and for any constant $\alpha$, there exist $\sigma_i \in \{-1, 1\}$ such that $\left| \sum_{i \in [k]} w_i \sigma_i - \alpha \right| = O(\sqrt{k}/2^k)$. To prove Theorem 3, first observe that a data point pays if the weighed sum of its neighbors is opposite in sign but smaller in magnitude compared to its own weight. Our proof begins by showing that for any sampled feature vector $x$, there is positive probability independent of the dataset size that it falls within a region with label noise, and its $k$ neighbors fall within this region as well. Due to label noise, there is positive probability that the neighbor labels attain the bound guaranteed by probabilistic optimal partitioning. Jointly, the lower bounds show that any sampled feature vector $x$ pays a non-negative amount in expectation, and thus the overall sum of payments grows linearly with $m$.

Finally, combining Theorems 2 and 3, we obtain total payment discrepancy under mild regularity and label noise:

**Corollary 2** (Asymptotic discrepancy; Informal)**.** *For distributions with bounded features and valuations, non-trivial signal and some regions with some label noise, it holds that $\mathrm{pay}_m(\mathcal{A}_{\mathrm{lin}}) = O(1)$ whereas $\mathrm{pay}_m(\mathcal{A}_{k\text{-NN}}) = \Omega(m)$.*

In this context, we also note that $L_2$-regularized linear classifiers have bounded classification stability (Bousquet & Elisseeff, 2002), while $k$-NN classifiers satisfy a weaker notion of hypothesis stability (Devroye & Wagner, 1979).

**Algorithm 1** `ComputeAllPaymentsExact`

---

**Input:** Data $S = \{(x_i, y_i)\}_{i=1}^m$, bids $\boldsymbol{b}$, $\beta$-class. stable objective $\mathcal{O}(h; \boldsymbol{w})$, constant $c \geq \frac{\kappa^2 \sigma}{2\lambda}$, toler. $\tau$, apx. $\epsilon$

1: initialize $\boldsymbol{p} = \boldsymbol{0}$
2: solve $\hat{f} = \operatorname{argmin}_{f \in F} \mathcal{O}(h_f; \boldsymbol{b})$ and set $\hat{h} = h_{\hat{f}}$
3: $\mu_i \leftarrow y_i \hat{f}(x_i) \quad \forall i \in S$    ▷ *compute margins*
4: $\beta_i \leftarrow c \cdot b_i \quad \forall i \in S$    ▷ *relevance thresholds*
5: **for** $i = 1, \ldots, m$ **do**
6:   **if** $\mu_i \in [0, \beta_i + \tau]$ **then**
7:    solve $\hat{h}_0 = \operatorname{argmin}_{h \in H} \mathcal{O}(h; (0, \boldsymbol{b}_{-i}))$
8:    **if** $a_i(\hat{h}_0) = 0$ **then**
9:     $p_i \leftarrow \texttt{BinarySearch}(\mathcal{O}(h; (\cdot, \boldsymbol{b}_{-i})), [0, b_i], \epsilon)$
10: **return** $\boldsymbol{p} = (p_1, \ldots, p_m)$

---

The question of whether notions of algorithmic stability can directly characterize the induced auction welfare is an intriguing direction for future inquiry.

## 6. Algorithms

We now turn to the question of how to compute payments. By Eq. (7), payments require solving a nested optimization problem, since the allocation $a_i(z; \boldsymbol{b}_{-i})$ in the constraints is determined by the learned classifier $\hat{h}$, which in itself is the argmin of $\mathcal{O}(z, \boldsymbol{b}_{-i})$ (Eq. (5)). Solving such problems directly is notoriously hard, despite the simple (linear) objective, and even when the nested learning objective is convex[1]. We therefore resort to other techniques for computing user payments, and give several alternative algorithms differing in how they balance runtime with probabilistic guarantees. The algorithms we present apply to any computable learning objective $\mathcal{O}$. However, and leveraging our results from Sec. 5, we show that stable algorithms also provide significant runtime improvements.

**Efficient numerical approximation.** Consider first a single user $i \in [m]$. Since $a_i$ is both binary and monotone in $z$, it is in effect a step function whose threshold lies at the critical bid, i.e., $a_i(z; \boldsymbol{b}_{-i}) = \mathbb{1}\{z \geq b_i^c\}$. Computing $b_i^c$ therefore amounts to finding the threshold of $a_i$. Since the domain of $z$ is bounded to $[0, b_i]$, this can be done efficiently using binary search, which guarantees an $\epsilon$-approximation of $b_i^c$ with $k \geq \lceil \log_2 \frac{b_i}{\epsilon} \rceil$ search steps for any $\epsilon$.

The above procedure can be repeated for all points to obtain the set of critical bids $b_1^c, \ldots, b_m^c$. However, since each step in each search requires solving a learning problem, this can be quite prohibitive. Luckily, classification-stable learning algorithms permit a drastic reduction in effective runtime

---

[1] See Dempe (2002). We note that payment computation for $k$-NN classifiers is simpler due to their localized nature, and given by a closed form formula. See Appendix A.8.

*Table 1.* **Comparison of payment algorithms.** $\mathcal{O}$ is the learning objective, $\widetilde{m}$ the number of candidate points, $k$ the number of search steps, $N$ the number of (possible) random draws. Typically $N \ll k$, and we expect $\widetilde{m} \ll m$. For SVM in particular, $\widetilde{m}$ is constant, and calls to $\mathcal{O}$ require only a subset of data points.

| | algorithm | calls to $\mathcal{O}$ | DSIC | IR | $p_i = b_i^c$ |
|---|---|---|---|---|---|
| stable | search | $1 + \widetilde{m}k$ | ✓ | ✓ | ✓ |
| | random | $1 + \widetilde{m}N$ | $\mathbb{E}$ | ✓ | $\mathbb{E}$ |
| | one-shot | $N$ | ✓ | ✓ | ✗ |
| non-stable | | $1 + mk$ | ✓ | ✓ | ✓ |

without compromising guarantees. First, Lemma 2 implies that only points with $y_i f(x_i) \in [0, \beta_i]$, where $f$ is trained with $\mathcal{O}(\boldsymbol{b})$, *might* require computation; all other points get $p_i = 0$ by definition. This reduces the number of candidate points possibly requiring search to $\widetilde{m} < m$. For these candidate points, it suffices to first compute $a_i(0; \boldsymbol{b}_i)$, and proceed only if the result is 0 (since otherwise, no other $z \in [0, b_i]$ will flip the allocation). In our experiments, the vast majority of candidate points terminate at this step, and so do not require search. For linear classifiers, Thm. 2 then ensures that $\widetilde{m}$ is constant and does not grow with $m$. The result is that a full search is needed only for a small subset of all points. Pseudocode for the full procedure is given in Algorithm 1.

For the SVM algorithm in particular, stability can further help reduce the runtime of each call to $\mathcal{O}_{\text{SVM}}(z; \boldsymbol{b}_{-i})$.

**Lemma 3.** *Let $f$ be the score function learned from $\mathcal{O}_{\text{SVM}}(\boldsymbol{b})$, and consider some user $i$. For all $j \in [m]$ define $b_j^{(i)} = b_j$ if $y_j f(x_j) \leq 1 + 2\beta_i$ and 0 otherwise. Then $a_i(z; \boldsymbol{b}_{-i}) = a_i(z; \boldsymbol{b}_{-i}^{(i)})$ for all $z$, and $p_i(\boldsymbol{b}) = p_i(\boldsymbol{b}^{(i)})$.*

Proof in Appendix B.1. Lemma 3 implies that all points $j$ with $b_j' = 0$ can effectively be removed from $S$ for the purpose of computing prices and allocations. As such points typically comprise the majority, this has potential to significantly reduce the runtime of each call to $\mathcal{O}$. In practice and for simplicity we replace $\beta_i$ in the condition with the global $\beta$, and for numerical stability add a small tolerance $\tau$.

**Randomized algorithm.** One way to further reduce runtime, and in particular the number of calls to $\mathcal{O}$ per candidate point, is via randomization. Consider a candidate user $i$, i.e., one whose critical bid $b_i^c$ requires explicit computation. As observed by Archer et al. (2004), $b_i^c$ can also be expressed as the expected (dis)allocation of bids $z$ drawn at random from the uniform distribution over $[0, b_i]$:

$$b_i^c = \mathbb{E}_{z \sim U(0, b_i)}[1 - a(z; \boldsymbol{b}_{-i})] \tag{10}$$

This follows since, as a step function with threshold at $b_i^c$, the integral over $a_i(z; \boldsymbol{b}_{-i})$ in the range $z \in [0, b_i]$ is precisely $1 - b_i^c$. The implication is that estimating $b_i^c$ by computing

$\mathcal{O}(z; \boldsymbol{b}_{-i})$ for a *single* $z \sim U(0, b_i)$ guarantees an expected payment of $p(b_i; \boldsymbol{b}_{-i})$ for user $i$, and so maintains expected truthful reporting when users maximize expected utility; see Algo. 3 in Appendix B. This approach requires a single call to $\mathcal{O}$, rather than $O(\log(1/\epsilon))$ as with line search. For stable algorithms, $b_i$ can be replaced with $\beta_i$ as before.

The downside is that randomization comes at the cost of variance in payments for users. In some sense, this 'shifts' compute costs incurred by the system to uncertainty in payments imposed on users. A better balance can be attained by drawing $N$ random bids and averaging, which maintains expectation while reducing variance by $1/\sqrt{N}$. The overhead is the need for $N$ calls to $\mathcal{O}$ per candidate user rather than one, though as before, the number of such users is typically very small.

**One-shot payments.** A final alternative is to compute payments for all users together using a *single* call to $\mathcal{O}$, but at the cost of both higher variance and a possible loss in expected welfare. Using the method of Babaioff et al. (2015), users are randomly split into 'paying' and 'rebate' groups. Paying users are charged the maximal $p_i = b_i$ if $a_i = 1$, whereas rebate users with $a_i = 1$ receive (random) negative payments. All users with $a_i = 0$ pay zero. The mechanism is IR and truthful per instance. The ratio of group sizes and randomness in payments are determined by global parameters controlling the tradeoff between bias (in welfare) and variance (in payments). Both can be reduced by averaging payments over $N$ trials. See Algo. 4 in Appendix B.

## 7. Experiments

In this section, we empirically explore our setting and approach. We begin with experiments focusing on payments using synthetic data, and then study the relation between welfare and accuracy using real and semi-synthetic data. In all experiments we use learning algorithms that comply with our upper bound, and the exact payment algorithm; see Appendix D for a comparison to the randomized algorithms and additional experiments. Results are averaged over multiple random trials. Full experimental details in Appendix C.

### 7.1. Payments in accuracy auctions

**Number of paying users.** Thm. 2 states that the number of paying users is bounded by a constant. Here we demonstrate this empirically on a simple classification task. We use multivariate Gaussian class-conditional distributions $P(x|y) = \mathcal{N}(\boldsymbol{\mu}_y, \sigma^2 I_d)$ with means $\boldsymbol{\mu}_y$ and isotropic covariance parameterized by $\sigma$, and set $P(y = 1) = P(y = 0) = 1/2$. We use $d = 16$, fix $\|\boldsymbol{\mu}_1 - \boldsymbol{\mu}_0\| = 0.5$, and for simplicity set $v_i = 1$ for all $i$. The variance parameter $\sigma$ allows us to control the density of points in the interval $[0, \beta]$; increasing $\sigma$ is expected to entail a larger number of candidate points, and

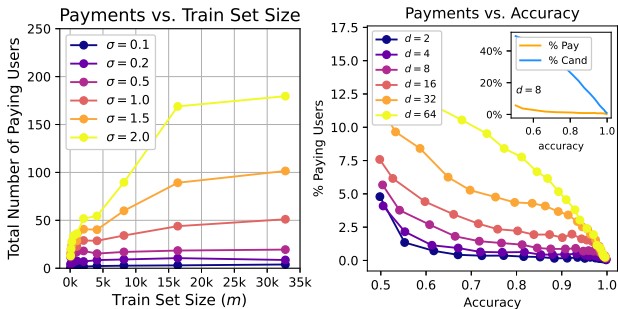

*Figure 2.* User payments under class-conditional multivariate Gaussian data. **(Left)** Number of paying users for increasing sample size and increasing variance ($\sigma$). **(Right)** Percentage of paying users as a function of model accuracy across varying class-mean distances and for increasing dimension ($d$).

likely more paying users. Fig. 2 (Left) shows the number of paying users $m_{\text{pay}}$ for increasing sample size $m$ across various $\sigma$. As can be seen, $m_{\text{pay}}$ plateaus quickly for all $\sigma$, where smaller $\sigma$ entails lower $m_{\text{pay}}$ and faster convergence. Though differences in $m_{\text{pay}}$ across $\sigma$ may appear large in absolute terms, these differences are negligible relative to the total sample size $m$, even for moderate values.

**Accuracy vs. payments.** In our previous experiments, increasing $\sigma$ also makes the data less separable, and hence reduces the maximal attainable accuracy. Here we aim to isolate and control the effect of accuracy on $m_{\text{pay}}$. We keep the same setup, but now use means $\boldsymbol{\mu}_y = (y\mu, 0, \dots, 0)$. This makes the classification task uni-dimensional by construction (i.e., only $x_1$ correlates with $y$), which allows us to control maximal accuracy by varying $\mu$ and with no dependence on $d$. We use $m = 500$ and fix $\sigma = 1$, set $\lambda = 1$, and control class-mean distances by varying $\mu \in [0, 6]$.

Fig. 2 (Right) shows the relation between accuracy and the number of payers $m_{\text{pay}}$ (out of $m$) for increasing $d$, where points on the plot are generated by varying $\mu \geq 0$. Results show that higher accuracy corresponds to lower payments across all $d$. This aligns with the idea that accuracy is a scarce resource, and the higher the scarcity (i.e., the lower the accuracy), the higher the cost of truthful elicitation. Note the trend is sharper trend for larger $d$. The inset figure exemplifies for $d = 8$ the number of paying vs. candidate users our algorithm identifies using Lemma 2. This suggests the constants in our upper bound are far from tight for this data.

### 7.2. Welfare, accuracy, and payments

To demonstrate the relation between welfare and accuracy we use the publicly available `folktables` dataset (Ding et al., 2021), and focus on the ACSIncome task. Values $v_i$ are set either according to the WKHP (usual hours worked per week past 12 months) attribute, denoted $z$ and which is not used in $x$, or to correlate with $y$. We use flexible

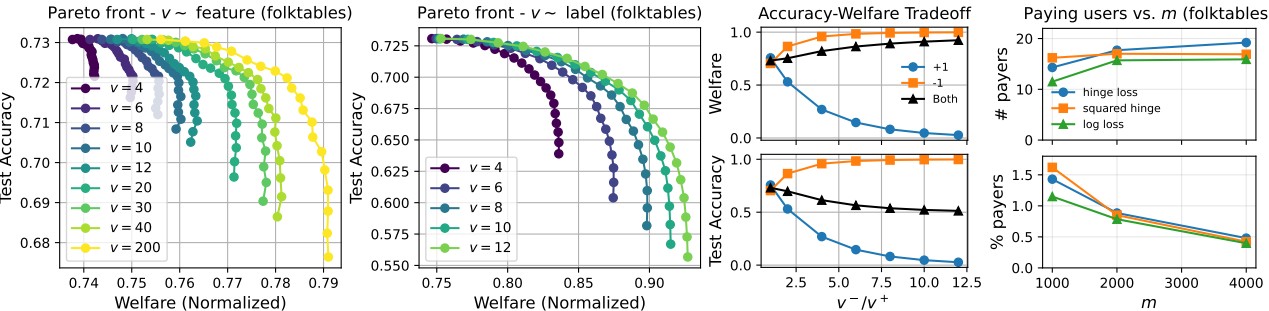

*Figure 3.* Welfare, accuracy, and payments on the `folktables` dataset. **(Left)** Tradeoff in accuracy vs. welfare under feature-based values for increasing valuation dispersion (controlled by $v_+$). **(Center-left)** Tradeoff in accuracy vs. welfare under label-based values for increasing valuation gap (controlled by $v_+$). **(Center-right)** For label-based values, per-class and overall welfare (top) and accuracy (bottom) when optimizing welfare ($\alpha = 1$). **(Right)** For label-based values, number (top) and percentage (bottom) of paying users under different losses complying with Thm. 2 for increasing sample size $m$.

mappings $v(z)$ and $v(y)$ allowing to control the variance of values via a parameter $v_+$. Full details in Appendix C.2.

**Accuracy-welfare tradeoffs.** Our first question considers the possible welfare gap due to naïvely optimizing (unweighted) accuracy vs. optimizing welfare directly. To this end, we interpolate between these two extremes by training with example weights $w_i = (1 - \alpha) + \alpha v_i$ for a range of $\alpha \in [0, 1]$, where $\alpha = 0$ recovers the accuracy objective and $\alpha = 1$ recovers welfare. This simulates a setting where the system benefits from maximizing accuracy, but also cares about welfare, whether endogenously or due to regulation.

Fig. 3 (Left) shows Pareto curves comparing welfare and accuracy of learned models across values of $\alpha$, and for varying degrees of dispersion of $v(z)$. As can be seen, welfare improves by up to $5\%$ when it is explicitly optimized. This becomes more distinct when the variance in $v$ is larger and weights are farther from uniform. Note that even for large dispersion, the curve is efficient in that welfare can be significantly improved with relatively little loss in accuracy.

**Allocation.** To better understand how welfare improves, we examine a more controlled setting where $v = v(y)$, allowing us to examine how allocations from predictions shift between user groups. Specifically we set $v(0) = 1$ and $v(1) = v_+$, and vary $v_+$. Fig.3 (Center-left) shows that the resulting Pareto curves follow generally similar trends, albeit with much larger gaps in welfare ($+23\%$) and accuracy ($-21\%$). Fig.3 (Center-right) compares accuracy and welfare for users from each class and overall under the welfare objective ($\alpha = 1$). As the gap in values between groups increases ($v_+$), the number of correct predictions for positive (high-value) users increases, and for negative (low-value) users decreases. This naturally results in increasing overall welfare, but also in lower accuracy. Thus, using true values as example weights in the objective results in less, but better, allocations.

**Payments.** Fig. 3 (Right) shows the number (top) and proportion (bottom) of paying users for increasing samples size $m$ across different loss functions: hinge, squared hinge, and log-loss, all of which admit the bound of Thm. 2, though with different constants. To accommodate for the effect of sample size on generalization, we set $\lambda = 10^3/\sqrt{m}$, and verify that this controls overfitting across all experimental conditions. We observe that the number of paying users plateaus as $m$ increases, and remains generally very low (at most 20). Similarly, proportion decays from $1.5\%$ towards zero as $m$ increases. In Appendix D.4, we show this trend holds broadly across additional regularization schemes.

## 8. Discussion

This work is motivated by the idea that machine learning can, and should, be used to improve social welfare. But while some aspects of this goal—such as optimization and generalization—are generally well-studied, the economic aspects of this challenge remain underexplored. Here we focus on the particular challenge that arises from the information gap due to user valuations being private, and propose accuracy auctions as a possible approach for mitigating this gap. Our results suggest that, for linear classification stable learning algorithms, the cost of welfare is quite low, given that the sum of payments is bounded by a constant.

An interesting future question is whether this holds also for some non-stable algorithms (e.g., decision trees or deep neural networks). Another is whether the need for a payment-based mechanism can be relaxed, or even circumvented altogether. Importantly, private information is but one way in which human agency, coupled with limited resources, introduces economic considerations into the learning objective. We hope our work will motivate further discussion on how to design learning algorithms that maximize welfare in face of broader economic considerations.

## Impact Statement

Our work introduces a novel learning setting where the goal is to maximize welfare by eliciting truthful reports of private user values. To this end, and to enable tractable analysis, we make several key assumptions on both the setup and user behavior. First, we focus on cardinal welfare, and in particular a utilitarian formulation. Second, we assume that user utility derives from individual prediction accuracy, with correct and incorrect predictions associated with fixed and predetermined individualized values. Third, we assume that users are rational: they know their valuations, and act to maximize their utility. These assumptions are crucial for guaranteeing that our proposed accuracy auction is incentive-compatible, and the basis for requiring individual rationality. Looking forward, we hope future work relaxes these assumptions by considering broader social welfare functions and more expressive utility models.

While the type of assumptions we make are common in economics, it is well known that real users are rarely fully rational. Human-subject experiments show that this holds even for truthful auctions, and even when users are aware that truthful reporting is, by design, the optimal strategy (e.g., Noti et al., 2014). Nonetheless, we are hopeful that in our setting, incentive compatibility will still encourage truthful reporting, perhaps due to the small number of users who ultimately pay. We also hope that our method provides some robustness to boundedly rational behavior, but rigorous guarantees for such settings requires additional future work.

Maximizing welfare through our framework requires a capacity to collect payments from users, either directly or indirectly through utility reduction. This may not always be feasible, socially desirable, or—in some cases—ethical. For example, if payments are monetary, some users may lack the financial means to submit bids matching their true values, raising potential socio-economic fairness concern. Another concern is that, because payments depend on features, whether a given user pays may correlate—or even result from—membership in a particular social group. Thus, deciding whether or not to use accuracy auctions in a given domain requires careful deliberation. In particular, it requires determining whether payments are appropriate and, if so, which implementation is most suitable.

Ideally, a policymaker should weigh the risks and benefits of a proposed approach, compare it with other available approaches, and choose the one that best serves the public interest. For policymakers seeking to promote welfare using machine learning, accuracy auctions offer an alternative to an existing prevalent and widespread default approach, which is to maximize accuracy. As we argue, maximizing accuracy—by ignoring the private valuations and implicitly assuming uniform values—can lead to very low expected welfare. Our approach offers policymakers a choice: maxi-mize population accuracy without payments and hope that welfare is reasonable, or maximize population welfare while incurring the social cost of having some train-time users pay (in some form). Importantly, both approaches make some assumptions on human behavior and agency: accuracy maximization assumes them away, while accuracy auctions model them explicitly. Each carries its own risks.

Myerson's lemma implies that payments are necessary for maximizing welfare via truthful elicitation when users are rational (Hartline & Roughgarden, 2008). In this context, and from a policy perspective, our results are encouraging: only train-time users (possibly) pay; payments need not be monetary; linear classification-stable algorithms guarantee a constant number of paying users in theory; and simulated results suggest that empirically often only a handful do. Thus, even in domains where payments are undesirable, their extent is likely limited. The flip side, however, is that limited payments can create inequity across users: a small minority pays to enable high welfare for all. This can be seen as a form of free riding. One mitigation strategy is for the system to subsidize payments, rather than collect them directly from users. We hope our work motivates the development of future approaches that relax or eliminate payments, though doing so will require circumventing truthful elicitation, and may require relying on different, perhaps stronger, assumptions.

## Acknowledgements

The authors would like to thank Axel Niemeyer and anonymous reviewers for their insightful remarks and valuable suggestions. Bana Sadi is supported by the Israel Council for Higher Education PBC scholarship for M.Sc. students in data science. Eden Saig is supported by the Israel Council for Higher Education PBC scholarship for Ph.D. students in data science, and the Zuckerman STEM Leadership Program. Nir Rosenfeld is supported by the Israel Science Foundation grant no. 278/22.

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

# A. Deferred Proofs

## A.1. Preliminaries

For the proofs, we denote the space of features by $\mathcal{X} = \mathbb{R}^d$, the space of binary labels by $\mathcal{Y} = \{-1, 1\}$, the class of hypotheses by $\mathcal{H} \subseteq \mathcal{Y}^{\mathcal{X}}$, and the space of valuations by $\mathcal{V} = [0, V]$. Denote the joint valuation-feature-label-valuation space by $\mathcal{Z} = \mathcal{V} \times \mathcal{X} \times \mathcal{Y}$, and let $D \in \Delta(\mathcal{Z})$ be a distribution over feature-label-valuation triplets. We denote the training set size by $n$, and a training set drawn i.i.d. from the joint distribution by $S = \{(w_i, \boldsymbol{x}_i, y_i)\}_{i=1}^m \sim D^n$. A sample-weighted learning algorithm is a mapping $\mathcal{A} : \mathcal{Z}^* \to \mathcal{H}$, where $\mathcal{Z}^*$ is the set of finite sequences from $\mathcal{Z}$, formally $\mathcal{Z}^* = \bigcup_{k=0}^\infty \mathcal{Z}^k$. We denote by $h_S(x)$ the classifier learned using training set $S$. We denote inner products by $\langle \cdot, \cdot \rangle$, and vector norms by $\|\cdot\|$. To simplify notation, we assume the norm is $L_2$ unless stated otherwise.

## A.2. Asymmetric Valuations

In Section 3, we implicitly assume that bidding agents associate accurate prediction with value $v$, and inaccurate predictions with value $0$. However, in practical applications, the agent may assign non-zero utility to inaccurate predictions. Here we show that our auction model directly extends to settings with non-zero valuation for inaccurate predictions.

Formally, for a given agent, denote by $v^+ \in \mathbb{R}$ their valuation of an accurate prediction, and denote by $v^- < v^+$ their valuation of an inaccurate prediction. The pseudo-linear utility of each agent, given in Section 3 by Equation (1), thus generalizes to:

$$u(y, \hat{y}; v^+, v^-) = v^+ \cdot \mathbb{1}\{y = \hat{y}\} + v^- \cdot \mathbb{1}\{y \neq \hat{y}\}$$
$$= (v^+ - v^-) \cdot \mathbb{1}\{y = \hat{y}\} + v^-$$

This utility is pseudo-linear in the difference $(v^+ - v^-) > 0$, and shifted by an agent-dependent constant $v^-$ that does not depend on the allocation or payment. Since dominant-strategy equilibria are invariant to constant shifts in payoffs, any auction that is DSIC with respect to valuations $\tilde{v} = (v^+ - v^-)$ of accurate predictions and valuation of $0$ for inaccurate predictions will also be DSIC in the asymmetric valuations case. Asymmetric valuations therefore induce a *single-parameter* auction setup if the agents are asked to report their marginal difference, and the properties of the auction are identical to the properties of the auction defined in Section 3. In particular, we also note that welfare-maximizing allocation are also invariant to constant shifts in agent utility due to linearity of the welfare objective.

## A.3. Score-Based Classification

Score-based classifiers make predictions by thresholding a scalar score function. Formally, given a score function $s : \mathcal{X} \to \mathbb{R}$, a score-based classifier is a function of the form:

$$h(x) = \text{sign}(f(\boldsymbol{x})) \tag{11}$$

Common examples include linear classifiers corresponding to $f(\boldsymbol{x}) = \langle \boldsymbol{\theta}, \boldsymbol{x} \rangle$ for a feature vector $\boldsymbol{\theta}$ (discussed in Section A.6), weighted nearest neighbors corresponding to $f(\boldsymbol{x}) = \sum_{(w_i, \boldsymbol{x}_i, y_i) \in N_k(\boldsymbol{x})} w_i y_i$ (discussed in Section A.7), kernel support vector machines, and binary classification neural networks $\phi : \mathcal{X} \to \mathbb{R}^2$ corresponding to $f(\boldsymbol{x}) = \phi_1(\boldsymbol{x}) - \phi_0(\boldsymbol{x})$.

## A.4. Score-Based Empirical Risk Minimization

Score-based empirical risk minimization learning algorithms optimize empirical loss over a class of score functions $\mathcal{F} \subseteq \mathbb{R}^{\mathcal{X}}$. Given a score function $f$ and feature-label pair $(\boldsymbol{x}, y)$, we denote the loss function by $\ell(t)$, where $t = \ell(f(\boldsymbol{x}) \cdot y) \in \mathbb{R}$. For score-based classifiers as defined above, we assume that the margin-based loss $\ell$ is monotonically decreasing with $t$. Note that this captures the common binary classification loss functions, including the 0-1 loss $\ell_{01}(t) = \mathbb{I}[t \leq 0]$, the hinge loss $\ell_{\text{hinge}}(t) = \max(0, 1 - t)$ and the logistic loss. With these definitions, we formally define the score-based learning rules:

**Definition 2** (Empirical regularized classification score loss)**.** Let $s \in \mathcal{F}$ be a score function, let $\{(w_i, \boldsymbol{x}_i, y_i)\}_{i=1}^m \in \mathcal{Z}^m$ be a weighted training set, let $\ell : \mathbb{R} \to \mathbb{R}$ be a loss function, and let $R : \mathcal{F} \to \mathbb{R}$ be a regularizer. The empirical regularized classification score loss of $f$ is:

$$L(f) = \frac{1}{m} \sum_{i=1}^m w_i \ell\left(f(\boldsymbol{x}_i) \cdot y_i\right) + \lambda R(f) \tag{12}$$

**Definition 3** (Score-based empirical risk minimization algorithm). Let $\mathcal{F}$ be a class of score functions, and let $L$ be a classification loss as defined above. A score-based learning algorithm outputs a loss-minimizing score function:

$$f^* \in \underset{f \in \mathcal{F}}{\operatorname{argmin}} L(f) \tag{13}$$

In the context of accuracy auctions, weights in Equation (12) are given by bids (as in Equation (5)), $w_i = b_i$. We denote the learned score function by $f^*(x; \boldsymbol{b})$, and the allocation rule induced by the learning algorithm is:

$$a_i(\boldsymbol{b}) = \mathbb{I}\left[\operatorname{sign}\left(f^*(\boldsymbol{x}; \boldsymbol{b})\right) = y\right] \tag{14}$$

### A.5. Monotonicity of Score-Based Regularized Empirical Risk Minimization

*Proof of Theorem 1.* Consider a score-based learning algorithm with a monotonically-decreasing score function $\ell(t)$. Fix a non-weighted training set $\{(\boldsymbol{x}_i, y_i)\}_{i=1}^m$, fix an arbitrary coordinate $i \in [m]$, and fix the bids of all bidders except for agent $i$, denoted by $\boldsymbol{b}_{-i}$. Let $s \in \mathcal{F}$ be a score function, and denote by $L(b_i; f)$ the classification score loss of score function $f$ when the bids profile is $(b_i, \boldsymbol{b}_{-i})$ (see Definition 2). By Equation (12), observe that the loss $L(b_i; f)$ is a linear function of $b_i$, with slope given by $\frac{1}{m}\ell\left(f(\boldsymbol{x}_i) \cdot y_i\right)$, and offset $c = \frac{1}{m}\sum_{i' \neq i} b_{i'}\ell((f(\boldsymbol{x}_{i'}) \cdot y_{i'}) + \lambda R(f)$.

For each $b_i$, denote the loss of the optimal score function by $L^*(b_i) = \min_{f \in \mathcal{F}} L(b_i; f)$. The loss of the optimal score $L^*(b_i)$ is a minimum over functions linear in $b_i$, and is therefore concave. Denote by $f^*(\boldsymbol{x}_i; b_i)$ the score function learned by the algorithm, as given by Equation (13), and note that the slope at $b_i$ is $\ell\left(f^*(\boldsymbol{x}_i; b_i) \cdot y_i\right)$ by definition. Since the slope of concave functions is monotonically non-decreasing, the slope of $L^*(b_i)$ is monotonically non-increasing in $b_i$. Furthermore, since the loss function $\ell$ is decreasing, it therefore holds that the product $f^*(\boldsymbol{x}_i; b_i) \cdot y_i$ is non-decreasing with $b_i$. Finally, as $y_i$ is fixed, the score function $f^*(\boldsymbol{x}_i; b_i)$ is non-decreasing with $b_i$ for $y_i = 1$, and non-increasing with $b_i$ for $y_i = -1$, and therefore the accuracy allocation rule induced by the score-based algorithm is monotone as required. □

*Remark* (Extension of Theorem 1 to a wider class of threshold-value loss functions). In the proof of Theorem 1, we assume that $\ell(t)$ is monotonically decreasing with $t$, as it captures the common loss functions used in theory and practice, such as the 0-1, hinge, and binary cross-entropy loss functions. However, we note that a similar proof technique also applies for a wider family of loss functions, namely those for which there exist a separating value $\alpha \in \mathbb{R}$ such that $\ell(t) \geq \alpha$ for all $t \leq 0$, and $\ell(t) \leq \alpha$ for all $t > 0$. Observe that this class of functions captures monotone functions as well. To prove this, let $i \in [n]$, and assume without loss of generality that $y_i = 1$. The proof of Theorem 1 shows that $\ell(f^*(x_i; b_i) \cdot y_i) = \ell(f^*(x_i; b_i))$ is non-increasing in $b_i$, and therefore the loss $\ell(f^*(x_i; b_i))$ may cross the threshold value $\alpha$ at most once. If a crossing point does not exist, then the accuracy allocation is constant and trivially monotone. Otherwise, denote the crossing point by $b_i^*$. By definition it holds that $f^*(x_i; b_i) \leq 0$ for all $b_i \leq b_i^*$, and $f^*(x_i; b_i) \geq 0$ for all $b_i > b_i^*$, and therefore the accuracy allocation is monotone as required.

*Remark* (Allocation monotonicity in multiclass classification). In multi-class settings, the set of labels $\mathcal{Y}$ is discrete with cardinality possibly larger than 2, and score-based classifiers are commonly defined as $h(\boldsymbol{x}) = \operatorname{argmax}_{y \in \mathcal{Y}} f_y(\boldsymbol{x})$. The proof of Theorem 1 relies on the assumption that a point is classified correctly if and only if its loss is below a certain threshold. In a multi-class setting, this property which holds for the multi-class 0-1 loss, but does not always hold for smooth loss functions such as multi-class negative log likelihood (NLL). To illustrate this, consider a three-class setting with two probabilistic classifiers and two data points. Assume the labels are $y_1 = 1$, $y_2 = 2$, and the score are given by:

$$\begin{aligned} f_1(\boldsymbol{x}_1) &= (1, 0, 0) & f_1(\boldsymbol{x}_2) &= (0.3, 0.4, 0.3) \\ f_2(\boldsymbol{x}_1) &= (0.01, 0.99, 0) & f_2(\boldsymbol{x}_2) &= (0, 0.45, 0.55) \end{aligned}$$

Consider the NLL loss $-\log(p_y)$. For $b_2 = 0$, NLL is minimized by $f_1$ which predicts $\boldsymbol{x}_2$ correctly, but for sufficiently large $b_2$ we learn $f_2$ which predicts $x_2$ incorrectly, and the allocation is not monotone. Guaranteeing monotonicity in multi-class accuracy allocations is an intriguing direction for future work.

### A.6. Total Payments and Stability in $L_2$-regularized Linear Classification

#### A.6.1. REGULARIZED LINEAR CLASSIFICATION PRELIMINARIES

In this section, we focus on linear classifiers $h(\boldsymbol{x}) = \operatorname{sign}\left(\langle \boldsymbol{\theta}, \boldsymbol{x} \rangle\right)$ trained with admissible loss functions smooth near the origin, and $L_2$ regularization. Formally, given a dataset $S = \{(w_i, \boldsymbol{x}_i, y_i)\}_{i=1}^m$, a loss function $\ell(t)$, and a regularization

parameter $\lambda \geq 0$, the training loss is:

$$L(\boldsymbol{\theta}; S) = \frac{1}{m} \sum_{i=1}^{m} w_i \ell \left( \langle \boldsymbol{\theta}, \boldsymbol{x}_i \rangle \cdot y_i \right) + \lambda \|\boldsymbol{\theta}\|_2^2 \tag{15}$$

Here we assume that that $\ell(t)$ is convex and $\sigma$-Lipchitz with respect to $t = \langle \boldsymbol{\theta}, \boldsymbol{x} \rangle \cdot y$. For smoothness near the origin, we assume that $\ell(t)$ has a $K_\ell$-Lipchitz first derivative $\ell'(t)$ all $t \in (-1, 1)$. We denote the derivative of $\ell$ at the origin by $c_\ell = -\ell'(0)$, and focus on loss functions which satisfy $c_\ell > 0$. Loss functions satisfying these properties include:

- The hinge loss $\ell_{\mathrm{hinge}}(t) = \max\{0, 1 - t\}$ with Lipchitz constant $\sigma = 1$, derivative magnitude $c_\ell = -\ell'_{\mathrm{hinge}}(0) = 1$ at the origin, and first-derivative Lipchitz constant $K_\ell = 1$ for $t \in (-1, 1)$. Hinge loss is a canonical loss function for the Soft-SVM algorithm (Cortes & Vapnik, 1995) – See, e.g., Shalev-Shwartz & Ben-David (2014, Section 15.2).

- The log loss $\ell_{\log}(t) = \log(1 + \exp(-t))$ with constant $\sigma = 1$, $c_\ell = -\ell'_{\mathrm{hinge}}(0) = 0.5$, and first-derivative Lipchitz constant $K_\ell = 0.25$ for $t \in (-1, 1)$. Log loss is the canonical loss for the Logistic Regression algorithm – See, e.g., Shalev-Shwartz & Ben-David (2014, Section 9.3).

- The squared hinge $\ell_{\mathrm{hinge}}^2(t)$ for bounded data and features, with $\sigma$ depending on the data radius, $c_\ell = 2$, and $K_\ell = 2$.

**Learned classifier.** We denote the learned feature vector by:

$$\boldsymbol{\theta}_S = \operatorname*{argmin}_{\boldsymbol{\theta}} L(\boldsymbol{\theta}; S)$$

and the corresponding decision function by:

$$f_S(\boldsymbol{x}) = \langle \boldsymbol{\theta}_S, \boldsymbol{x} \rangle$$

Given distribution $D$, we denote the classifier learned in the population-limit ($n \to \infty$) by $\boldsymbol{\theta}_D$, and assume that $\boldsymbol{\theta}_D \neq 0$.

**Aggregate loss gradient.** In regions where $\ell(t)$ is differentiable, the gradient components of $L(\boldsymbol{\theta}; S)$ are given by:

$$(\nabla_{\boldsymbol{\theta}} L)_k = \frac{\partial L}{\partial \theta_k} = \frac{1}{m} \sum_{i=1}^{m} w_i \cdot \left( \left. \frac{\mathrm{d}\ell}{\mathrm{d}t} \right|_{t = \langle \boldsymbol{\theta}, \boldsymbol{x}_i \rangle \cdot y_i} \right) \cdot x_{i,k} \cdot y_i + 2\lambda \theta_k \tag{16}$$

A.6.2. SIGNAL STRENGTH

Our bounds depend on *weighed signal strength*, a data-dependent quantity which will be related the magnitude of the learned feature vector $\boldsymbol{\theta}_S$:

**Definition 4** (Weighted signal strength). Given a distribution $D \in \Delta(\mathcal{Z})$, the population signal strength is:

$$\begin{aligned} \boldsymbol{\mu}_D &= \mathbb{E}_{(\boldsymbol{x}, y, w) \sim D}[\boldsymbol{x} \cdot wy] \\ &= \mathbb{E}_{(w, \boldsymbol{x}, y) \sim D}[w\boldsymbol{x} \mid y = 1] \cdot \mathbb{P}[y = 1] - \mathbb{E}_{(w, \boldsymbol{x}, y) \sim D}[w\boldsymbol{x} \mid y = -1] \cdot \mathbb{P}[y = -1] \end{aligned}$$

given a training set $S \in \mathcal{Z}^m$, the empirical signal strength is defined analogously:

$$\boldsymbol{\mu}_S = \frac{1}{m} \sum_{i=1}^{m} \boldsymbol{x}_i \cdot w_i y_i$$

**Proposition 1.** *For any loss function $\ell(t)$ differentiable at the origin and any training set $S$, it holds that:*

$$\nabla_{\boldsymbol{\theta}} L(\boldsymbol{\theta}; S)|_{\boldsymbol{\theta} = 0} = -c_\ell \cdot \boldsymbol{\mu}_S$$

*Proof.* Evaluating Equation (16) at $\boldsymbol{\theta} = 0$, we obtain:

$$\left. \frac{\partial L}{\partial \theta_k} \right|_{\boldsymbol{\theta} = 0} = -c_\ell \frac{1}{m} \sum_{i=1}^{m} w_i x_{i,k} y_i = -c_\ell \cdot (\boldsymbol{\mu}_S)_k \tag{17}$$

and aggregating the different vector components yields $\nabla_{\boldsymbol{\theta}} L(\boldsymbol{\theta}; S)|_{\boldsymbol{\theta} = 0} = -c_\ell \cdot \boldsymbol{\mu}_S$ as required. $\square$

**Corollary 3.** $\boldsymbol{\theta}_S \neq 0$ *if and only if* $\boldsymbol{\mu}_S \neq 0$.

*Remark.* The analogous argument also holds in the population limit, yielding $\boldsymbol{\theta}_D \neq 0 \leftrightarrow \boldsymbol{\mu}_D \neq 0$.

**Lemma 4.** *Let* $\boldsymbol{\theta} \in \mathbb{R}^d$ *be a vector of weights with lower-bounded norm* $\|\boldsymbol{\theta}\| \geq B_{\boldsymbol{\theta}}$, *and let* $X \in \Delta(\mathcal{X})$ *be a distribution over features* $\boldsymbol{x} \sim X$ *with bounded support* $\|\boldsymbol{x}\| \leq B_x$ *and probability density bounded by* $M_x$. *Let* $a, b > 0$. *Then there exists a constant* $C = C(B_x, B_{\boldsymbol{\theta}}, M_x)$ *such that:*

$$\mathbb{P}_{\boldsymbol{x} \sim X}[\langle \boldsymbol{\theta}, \boldsymbol{x} \rangle \in [a, b]] \leq C(b - a)$$

*Proof.* By assumption, the density of $X$ is bounded by a constant $M_x > 0$. Define the function $g(\boldsymbol{x}) = M_x \mathbb{I}[\|\boldsymbol{x}\| \leq B_x]$, and note that $g(\boldsymbol{x})$ upper bounds the probability density of $X$ by definition. Denote the normalized model parameters by $\hat{\boldsymbol{\theta}}$, such that $\boldsymbol{\theta} = \hat{\boldsymbol{\theta}} \|\boldsymbol{\theta}\|$. It holds that:

$$
\begin{aligned}
\mathbb{P}_{\boldsymbol{x} \sim X}[\langle \boldsymbol{\theta}, \boldsymbol{x} \rangle \in [a, b]] &= \mathbb{P}_{\boldsymbol{x} \sim X}\left[\left\langle \hat{\boldsymbol{\theta}}, \boldsymbol{x} \right\rangle \in \left[\frac{a}{\|\boldsymbol{\theta}\|}, \frac{b}{\|\boldsymbol{\theta}\|}\right]\right] \\
&\leq \int_{\boldsymbol{x} \in \mathbb{R}^d} g(x) \mathbb{I}\left[\left\langle \hat{\boldsymbol{\theta}}, x \right\rangle \in \left[\frac{a}{\|\boldsymbol{\theta}\|}, \frac{b}{\|\boldsymbol{\theta}\|}\right]\right] \mathrm{d}\boldsymbol{x} \\
&\leq \frac{b - a}{\|\boldsymbol{\theta}\|} \cdot M_x \cdot B_d
\end{aligned}
$$

where $B_d$ is the volume of the $(d-1)$-dimensional ball of radius $B_w$. The final result is obtained by noting that $\|\boldsymbol{\theta}\| \geq B_w$ and setting $C = \frac{M_x \cdot B_d}{B_w}$. $\qquad\square$

**Lemma 5.** *Let* $D$ *be a distribution over* $d$-*dimensional vectors with bounded support* $\|\boldsymbol{v}\| \leq B$, *and let* $\boldsymbol{v}_1, \ldots, \boldsymbol{v}_m \sim D$. *Denote* $\boldsymbol{\mu}_D = \mathbb{E}_{\boldsymbol{v} \sim D}[\boldsymbol{v}]$ *and* $\boldsymbol{\mu}_S = \frac{1}{m} \sum_{i=1}^m \boldsymbol{v}_i$. *Then* $\|\boldsymbol{\mu}_S\| \geq \|\boldsymbol{\mu}\| / 2$ *with probability at least* $1 - \exp\left(-\frac{m\|\boldsymbol{\mu}_D\|^2}{8B^2}\right)$.

*Proof.* Let $\hat{\boldsymbol{\mu}}_D = \frac{\boldsymbol{\mu}_D}{\|\boldsymbol{\mu}_D\|}$. By the Cauchy-Schwarz inequality, we obtain:

$$\frac{1}{m} \sum_{i=1}^m \langle \hat{\boldsymbol{\mu}}, \boldsymbol{v}_i \rangle = \langle \hat{\boldsymbol{\mu}}, \boldsymbol{\mu}_S \rangle \leq \|\boldsymbol{\mu}_S\|$$

and therefore:

$$\mathbb{P}_D\left[\|\boldsymbol{\mu}_S\| \leq \frac{\|\boldsymbol{\mu}_D\|}{2}\right] \leq \mathbb{P}_D\left[\frac{1}{m} \sum_{i=1}^m \langle \hat{\boldsymbol{\mu}}, \boldsymbol{v}_i \rangle \leq \frac{\|\boldsymbol{\mu}_D\|}{2}\right]$$

By definition and linearity of expectation, it holds that $\mathbb{E}_{\boldsymbol{v}_i}[\langle \hat{\boldsymbol{\mu}}_D, \boldsymbol{v}_i \rangle] = \|\boldsymbol{\mu}_D\|$. Moreover, as $D$ has bounded support $\|\boldsymbol{v}\| \leq B$ and $\hat{\boldsymbol{\mu}}_D$ is normalized, it holds by Cauchy-Schwarz that $\langle \hat{\boldsymbol{\mu}}_D, \boldsymbol{v}_i \rangle \in [-B, B]$. We can therefore apply Hoeffding's inequality on the variables $\langle \hat{\boldsymbol{\mu}}_D, \boldsymbol{v} \rangle$ to obtain:

$$\mathbb{P}_D\left[\frac{1}{m} \sum_{i=1}^m \langle \hat{\boldsymbol{\mu}}, \boldsymbol{v}_i \rangle \leq \frac{\|\boldsymbol{\mu}_D\|}{2}\right] \leq \exp\left(-\frac{m\|\boldsymbol{\mu}_D\|^2}{8B^2}\right)$$

and hence:

$$\mathbb{P}_D\left[\|\boldsymbol{\mu}_S\| \geq \frac{\|\boldsymbol{\mu}_D\|}{2}\right] \geq 1 - \exp\left(-\frac{m\|\boldsymbol{\mu}_D\|^2}{8B^2}\right)$$

as required. $\qquad\square$

**Lemma 6.** *Consider a data distribution* $D$ *over a feature space* $\mathcal{Z}$ *such that* $\|\boldsymbol{x}\| \leq B$ *and* $w \leq W$, *with signal strength* $\boldsymbol{\mu}_D \neq 0$. *Consider a loss function satisfying the assumptions above, and characterized by constants* $c_\ell$ *and* $K_\ell$. *Fix a regularization parameter* $\lambda$. *Then there exists a constant* $C$ *such that for any training set* $S \sim D^m$, *it holds that:*

$$\|\boldsymbol{\theta}_S\| \geq \min\left\{\frac{1}{B}, \frac{c_\ell \cdot \|\boldsymbol{\mu}_D\|}{2(W \cdot K_\ell \cdot B^2 + 2\lambda)}\right\} \tag{18}$$

*with probability at least* $1 - \exp(-Cm)$.

*Proof.* Sample a training set $S \sim D^m$. We consider two cases: If the learned $\boldsymbol{\theta}_S$ satisfies $\|\boldsymbol{\theta}_S\| \geq 1/B$, the bound is satisfied trivially. Otherwise, it holds that $\|\boldsymbol{\theta}\| < 1/B$, and it also holds that $\|\boldsymbol{x}\| \leq B$ by assumption. Denote $t_i = \langle \boldsymbol{\theta}, \boldsymbol{x}_i \rangle \cdot y_i$. By Cauchy-Schwarz, it holds that:

$$|t_i| = |\langle \boldsymbol{\theta}, \boldsymbol{x}_i \rangle \cdot y_i| \leq \|\boldsymbol{\theta}\| \cdot \|\boldsymbol{x}_i\| < \frac{1}{B} \cdot B = 1$$

Therefore $|t_i| < 1$, and hence $t_i \in (-1, 1)$ for all $i \in [m]$. Thus, by Equation (16) and for $\boldsymbol{\theta}$ with magnitude upper bounded by $1/B$, each gradient component of the aggregate training loss, denoted by $(\nabla_{\boldsymbol{\theta}} L(\boldsymbol{\theta}; S))_k$ and given by Equation (16), is a sum of Lipschitz-continuous functions. The derivative of $\ell$ is $K_\ell$-Lipschitz by assumption, weights are bounded by $W$, features vectors $\boldsymbol{x}$ are bounded by $B$. Therefore, the loss gradient $\nabla_{\boldsymbol{\theta}} L$ is Lipschitz continuous with respect to $\boldsymbol{\theta}$ and the $L_2$ norm, with constant $K = W \cdot K_\ell \cdot B^2 + 2\lambda$. From the definition of Lipchitz continuity, we obtain:

$$\left\| \nabla_{\boldsymbol{\theta}} L|_{\boldsymbol{\theta}=0} - \nabla_{\boldsymbol{\theta}} L|_{\boldsymbol{\theta}=\boldsymbol{\theta}_S} \right\| \leq K \left\| 0 - \boldsymbol{\theta}_S \right\|$$

rearranging and applying Proposition 1, we obtain:

$$\|\boldsymbol{\theta}_S\| \geq \frac{1}{K} \cdot \left\| \nabla_{\boldsymbol{\theta}} L|_{\boldsymbol{\theta}=0} \right\| = \frac{c_\ell}{K} \cdot \|\boldsymbol{\mu}_S\|$$

By definition $\boldsymbol{\mu}_S = \frac{1}{m} \sum_{i=1}^m w_i \cdot \boldsymbol{x}_i \cdot y_i$, and by the boundedness assumption $\|w_i \cdot \boldsymbol{x}_i \cdot y_i\| \leq W \cdot B$. Thus, applying Lemma 5 we obtain that:

$$\|\boldsymbol{\theta}_S\| \geq \frac{c_\ell \cdot \|\boldsymbol{\mu}_D\|}{2K}$$

with probability at least $1 - \exp\left( -\frac{m\|\boldsymbol{\mu}_D\|}{8B^2 W^2} \right)$. Combining with the converse case $\|\boldsymbol{\theta}\| \geq 1/B$ and setting $C = \frac{\|\boldsymbol{\mu}_D\|}{8B^2 W^2}$ as the probability constant yields the required result, and thus Equation (18) holds.

$\square$

### A.6.3. STABILITY PROPERTIES IN WEIGHTED CLASSIFICATION SETTINGS

For algorithmic stability, we extend the definitions and related lemmas in Bousquet & Elisseeff (2002) to the weighted setting by considering a sample-weighted training objective, and identifying sample removal with setting $w_i \leftarrow 0$. In a notational convention closer to Bousquet & Elisseeff (2002), we consider the following training objective:

$$L(f) = \frac{1}{m} \sum_{i=1}^m w_i \ell\left( f(\boldsymbol{x}_i), y_i \right) + \lambda R(f)$$

where $f \in \mathcal{F}$ is a score function assumed to be of the form $f(\boldsymbol{x}) = \langle \boldsymbol{\theta}, \boldsymbol{x} \rangle$ for linear classifiers, $\ell(f(\boldsymbol{x}), y) = \ell(\langle \boldsymbol{\theta}, \boldsymbol{x} \rangle \cdot y)$ is a loss function, and $R(f)$ is a regularizer, assumed to be $R(f) = \|\boldsymbol{\theta}\|^2$ in the case of $L_2$ regularization. We recall the key definitions related to classification stability:

**Definition 5** (Classification stability; Bousquet & Elisseeff (2002, Definition 15)). Let $H$ be a collection of score-based classifiers $h(\boldsymbol{x}) = \text{sign}(f(\boldsymbol{x}))$ induced by score functions $\mathcal{F} \subseteq \mathbb{R}^{\mathcal{X}}$. Let $\mathcal{A}$ be a learning algorithm mapping sample sets $S$ of size $m$ to score functions $f \in \mathcal{F}$, denoted $f_S = \mathcal{A}(S)$. Then $\mathcal{A}$ is $\beta$-*classification stable* if:

$$\forall S \quad \forall i, j \in [m] \quad |f_S(\boldsymbol{x}_j) - f_{S^{\backslash i}}(\boldsymbol{x}_j)| \leq \beta \tag{19}$$

where $S^{\backslash i}$ denotes the sample set without the $i^{\text{th}}$ example, or equivalently setting $w_i \leftarrow 0$.

**Definition 6** ($\sigma$-admissibility; Bousquet & Elisseeff (2002, Definition 19)). A loss function $\ell(f(x), y)$ is $\sigma$-admissible if it is convex and $\sigma$-Lipschitz continuous with respect to its first argument:

$$\forall x, x' \in \mathcal{X}, y \in \mathcal{Y}: \quad |\ell(f(x), y) - \ell(f(x'), y)| \leq \sigma |f(x) - f(x')|$$

*Remark.* For classification tasks, we note that single-variable loss functions of the form $\ell(f(\boldsymbol{x}) \cdot y)$ as defined in Section A.6.1 are $\sigma$-admissible when $\ell(t)$ is $\sigma$-Lipschitz as a single-variable function.

The following lemma generalizes Bousquet & Elisseeff (2002, Lemma 20) to weighted risk minimization:

**Lemma 7** (Stability of $\sigma$-admissible regularized learning). *Let $\ell$ be $\sigma$-admissible loss function, and $R$ be a regularization functional defined on $\mathcal{F}$ such that for all training sets, the empirical loss minimization has a minimum in $\mathcal{F}$. Denote the minimizer of $L(f)$ by $f$, the minimizer of $L^{\backslash i}(f)$ by $f^{\backslash i}$, and $\Delta f = f^{\backslash i} - f$. For any $\tau \in [0, 1]$, it holds that:*

$$R(f) - R(f + \tau \Delta f) + R(f^{\backslash i}) - R(f^{\backslash i} - \tau \Delta f) \leq \frac{w_i \tau \sigma}{\lambda m} |\Delta f(x_i)| \tag{20}$$

*Proof.* Given a training set $S \in \mathcal{Z}^n$, let $i \in [n]$. Recall the score function loss in Definition 2, and denote:

$$L^{\backslash i}(f) = \frac{1}{m} \sum_{j \neq i} w_i \ell(f(x_i), y_i)$$

Denote $\Delta f = f - f^{\backslash i}$. For all $\tau \in [0, 1]$, any convex function $g$ satisfies:

$$g\left(x + t(y - x)\right) - g(x) \leq t\left(g(y) - g(x)\right)$$

Both $L$ and $L^{\backslash i}$ are convex as a function of their inputs as sums of convex functions. Therefore, for all $\tau \in [0, 1]$:

$$L^{\backslash i}\left(f + \tau \Delta f\right) - L^{\backslash i}\left(f\right) \leq \tau \left(L^{\backslash i}(f^{\backslash i}) - L^{\backslash i}(f)\right)$$

$$L^{\backslash i}\left(f^{\backslash i} - \tau \Delta f\right) - L^{\backslash i}\left(f^{\backslash i}\right) \leq \tau \left(L^{\backslash i}(f) - L^{\backslash i}(f^{\backslash i})\right)$$

Summing the two inequalities yields:

$$L^{\backslash i}\left(f + \tau \Delta f\right) - L^{\backslash i}\left(f\right) + L^{\backslash i}\left(f^{\backslash i} - \tau \Delta f\right) - L^{\backslash i}\left(f^{\backslash i}\right) \leq 0 \tag{21}$$

By the optimality of $f$ and $f^{\backslash i}$, we also have:

$$L(f) - L(f + \tau \Delta f) \leq 0$$

$$L^{\backslash i}(f^{\backslash i}) - L(f^{\backslash i} - \tau \Delta f) \leq 0$$

Summing the inequalities above and applying Equation (21), we obtain:

$$\frac{w_i}{m} \ell\left(f(x_i), y_i\right) - \frac{w_i}{m} \ell\left((f + \tau \Delta f)(x_i), y_i\right) + \lambda \left(R(f) - R(f + \tau \Delta f) + R(f^{\backslash i}) - R(f^{\backslash i} - \tau \Delta f)\right) \leq 0$$

and finally, by $\sigma$-admissibility, we obtain:

$$R(f) - R(f + \tau \Delta f) + R(f^{\backslash i}) - R(f^{\backslash i} - \tau \Delta f) \leq \frac{w_i \tau \sigma}{\lambda m} |\Delta f(x_i)|$$

as required. $\qquad\qquad\qquad\qquad\qquad\qquad\qquad\qquad\qquad\qquad\qquad\qquad\qquad\qquad\qquad\qquad\qquad\square$

The following lemma generalizes Bousquet & Elisseeff (2002, Example 2) to weighted risk minimization for $L^2$-regularized linear classifiers:

**Lemma 8** (Stability of $L^2$-regularized linear classifiers). *Consider a linear classification algorithm with $L_2$ regularization and a $\sigma$-admissible loss function, training on a dataset with bounded features $\|x\| \leq B$. For a weighted training objective such that $w_i \leq w_{\max}$, the classification stability of the algorithm satisfies:*

$$\beta(\boldsymbol{w}) \leq \frac{w_{\max} \sigma B}{2 \lambda m}$$

*Proof.* For $L_2$-regularized classifiers, it holds that $R(f) = \|f\|^2 = \langle f, f \rangle$, and therefore the LHS of Equation (20) given by Lemma 7 simplifies to:

$$\underbrace{R(f)}_{=\|f\|^2} - \underbrace{R(f + t\Delta f)}_{=\|f\|^2 + t^2\|\Delta f\|^2 + 2t\langle f, \Delta f\rangle} + \underbrace{R(f^{\backslash i})}_{=\|f^{\backslash i}\|^2} - \underbrace{R(f^{\backslash i} - t\Delta f)}_{=\|f\|^2 + t^2\|\Delta f\|^2 - 2t\langle f, \Delta f\rangle} = 2t\left\langle f^{\backslash i} - f, \Delta f \right\rangle - 2t^2 \|\Delta f\|^2$$

$$= 2t(1 - t)\|\Delta f\|^2$$

Thus Equation (20) is equivalent to:

$$2t(1-t)\|\Delta f\|^2 \leq \frac{w_i t \sigma}{\lambda m}|\Delta f(x_i)|$$

Dividing both sides by $t$ and taking $t \to 0$, we obtain:

$$\|\Delta f\|^2 \leq \frac{w_i \sigma}{2\lambda m}|\Delta f(x_i)| \tag{22}$$

By Cauchy-Schwarz, for $x_i$ it holds that:

$$|\Delta f(x_i)| = |\langle \Delta f, x_i \rangle| \leq \|\Delta f\| \cdot B \tag{23}$$

Multiplying Equation (22) by $B^2$, plugging Equation (23) and dividing by $|\Delta f(x_i)|$, we obtain:

$$|\Delta f(x_i)| \leq \frac{w_i \sigma B^2}{2\lambda m} \tag{24}$$

An upper bound on $|\Delta f(x_i)|$ implies classification stability by definition, and thus the classification stability satisfies:

$$\beta(\boldsymbol{w}) \leq w_{\max} \cdot \frac{\sigma B}{2\lambda m}$$

Also note that for the special case Soft-SVM with a linear kernel and uniform weights $w_i = 1$, the bound above coincides with Bousquet & Elisseeff (2002, Example 2). $\qquad\square$

*Remark* (Individual classification stability $\beta_i$). In the proof of Lemma 8, the bound established in Equation (24) can be interpreted as an *individual* classification stability bound: $\left|f^{\setminus i}(\boldsymbol{x}_i) - f(\boldsymbol{x}_i)\right| \leq \beta_i$, where $\beta_i = w_i \sigma B^2 / 2\lambda m$. We use this notion for constructing efficient payment calculation algorithms in Section 6.

### A.6.4. AUCTIONS OF $L_2$-REGULARIZED LINEAR CLASSIFIERS

**Proposition 2.** *In an auction induced by an algorithm with weighted classification stability $\beta$, agent $i$ may pay only if $f(x_i) \cdot y_i \in [0, \beta]$.*

*Proof.* We prove this claim by showing that agent $i$ does not pay when the condition does not hold. Denote the payment of agent $i$ by $p_i$, and assume without loss of generality that $y_i = 1$. We consider two cases. If $f(x_i) > \beta$, then $h(x_i) = y_i = 1$. By definition of classification stability (Definition 5), it holds that $f^{\setminus i}(x_i) \geq f(x_i) - \beta > 0$ and therefore the leave-one-out classifier retains the same prediction $h^{\setminus i}(x_i) = h(x_i) = y_i = 1$. Then, by Myerson's payment rule (Equation (7)), it holds that $p_i = 0$ as required. Otherwise, if $f(x_i) < 0$ then $h(x_i) \neq y_i$, and from monotonicity (Theorem 1) it holds that $h^{\setminus i}(x_i) = h(x_i) = 0$, and $p_i = 0$ by Myerson's payment rule as well. $\qquad\square$

**Theorem 4** (Formal statement of Theorem 2). *Consider an accuracy auction induced by a linear classification algorithm with $L_2$ regularization, bounded features $\|\boldsymbol{x}\| \leq B$, feature probability density bound $M$, bounded valuations $v \leq V$, expected weighted signal strength $\boldsymbol{\mu}_D$, and fixed regularization parameter $\lambda$. For a random dataset of size $m$, denote the number of paying users by $\#\text{pay}(S)$. The expected value of $\#\text{pay}(S)$ is bounded by a constant:*

$$\mathbb{E}_{S \sim D^m}[\#\text{pay}(S)] = O\left(\frac{VB^3 M}{\lambda \min\left\{\frac{1}{B}, \frac{c_\ell \cdot \|\boldsymbol{\mu}_D\|}{2(W \cdot K_\ell \cdot B^2 + 2\lambda)}\right\}}\right)$$

*Furthermore, for expected revenue it holds that:*

$$\mathbb{E}_{S \sim D^m}[\text{revenue}(S)] \leq V \mathbb{E}_{S \sim D^m}[\#\text{pay}(S)]$$

*Proof.* Denote the classification stability of the weighted algorithm by $\beta$, fix a training set and assume valuations are bounded by $V$. Denote the payment of agent $i$ by $p_i$. By definition, it holds that:

$$\#\text{pay}(S) = \sum_{i=1}^{m} \mathbb{I}[p_i > 0]$$

By Proposition 2, it holds that:

$$\sum_{i=1}^{m} \mathbb{I}\left[p_i > 0\right] \leq \sum_{i=1}^{m} \mathbb{I}\left[0 \leq f_S(\boldsymbol{x}_i) \cdot y_i \leq \beta\right]$$

Taking the expectation over training set $S \sim D^n$, we obtain from linearity of expectation:

$$\mathbb{E}_S[\#\mathrm{pay}(S)] \leq m \cdot \mathbb{P}_S[0 \leq f_S(\boldsymbol{x}_i) \cdot y_i \leq \beta]$$

where $i \in [m]$ here denotes the index of an arbitrary point $(\boldsymbol{x}_i, y_i)$ in the random training set $S$. Note that probability is identical for all $i \in [m]$ as data points in the training set are sampled i.i.d..

Next, we leverage classification stability once more. Adding and subtracting $f_{S\backslash i}(\boldsymbol{x}_i) \cdot y_i$, we observe that:

$$f_S(\boldsymbol{x}_i) \cdot y_i = f_{S\backslash i}(\boldsymbol{x}_i) \cdot y_i + \underbrace{(f_S(\boldsymbol{x}_i) - f_{S\backslash i}(\boldsymbol{x}_i)) \cdot y_i}_{\in [-\beta, \beta]}$$

Note that $|f(\boldsymbol{x}_i) - f_{S\backslash i}(\boldsymbol{x}_i)| \leq \beta$ by classification stability, and therefore:

$$\mathbb{E}_{S \sim D^m}[\#\mathrm{pay}(S)] \leq m \cdot \underbrace{\mathbb{P}_S[-\beta \leq f_{S\backslash i}(\boldsymbol{x}_i) \cdot y_i \leq 2\beta]}_{(*)} \tag{25}$$

Denote $(*) = m \cdot \mathbb{P}_S[-\beta \leq f_{S\backslash i}(\boldsymbol{x}_i) \cdot y_i \leq 2\beta]$. Applying the law of total probability, we obtain:

$$\begin{aligned}
(*) &= \mathbb{P}_S[-\beta \leq f_{S\backslash i}(\boldsymbol{x}_i) \cdot y_i \leq 2\beta] \\
&= \mathbb{E}_{S' \sim D^{m-1}}\left[\mathbb{P}_{(w,\boldsymbol{x},y)\sim D}[-\beta \leq f_{S'}(\boldsymbol{x}) \cdot y \leq 2\beta]\right] \\
&= \mathbb{E}_{S' \sim D^{m-1}}\left[\mathbb{E}_{(w,x,y)\sim D}[\mathbb{I}\left[\langle \boldsymbol{\theta}_{S'}, \boldsymbol{x}\rangle \cdot y \in [-\beta, 2\beta]\right]]\right]
\end{aligned}$$

Denote by $C_1 = \min\left\{\frac{1}{B}, \frac{c_\ell \cdot \|\boldsymbol{\mu}_D\|}{2(W \cdot K_\ell \cdot B^2 + 2\lambda)}\right\}$ the high-probability lower bound for $\|\boldsymbol{\theta}_{S'}\|$ given by Lemma 6, and by $C_2$ the corresponding exponential convergence rate constant. For a random sample of $S' \sim D^{m-1}$, define the event $A$ as $\|\boldsymbol{\theta}_{S'}\| \geq C_1$. From the law of total probability:

$$(*) = p(A)\mathbb{E}_{S'|A}\left[\mathbb{E}_{(w,x,y)\sim D}[\mathbb{I}\left[\langle \boldsymbol{\theta}_{S'}, \boldsymbol{x}\rangle \cdot y \in [-\beta, 2\beta]\right]]\right] + p(A^c)\mathbb{E}_{S'|A^c}\left[\mathbb{E}_{(w,x,y)\sim D}[\mathbb{I}\left[\langle \boldsymbol{\theta}_{S'}, \boldsymbol{x}\rangle \cdot y \in [-\beta, 2\beta]\right]]\right]$$

From Lemma 6, we obtain a bound on the probability of the complementary event $p(A^c) \leq 2d\exp\left(-C_2(m-1)\right)$. Additionally, from Lemma 4, there exist a constant $C_3$ depending additionally on the norm lower bound $C_1$, feature probability density bound $M$, and feature magnitude bound $B$, denoted $C_3 = \frac{MB}{C_1}$ such that:

$$\mathbb{E}_{(w,x,y)\sim D}[\mathbb{I}\left[\langle \boldsymbol{\theta}_{S'}, \boldsymbol{x}\rangle \cdot y \in [-\beta, 2\beta]\right]] \leq C_3 \cdot 3\beta$$

Jointly, these yield an upper bound on $(*)$:

$$(*) \leq 3C_3\beta + 2d\exp\left(-C_2(m-1)\right)$$

By Lemma 8, it holds that $\beta \leq \frac{w_{\max}\kappa^2}{2\lambda m} \leq \frac{VB^2}{2\lambda m}$, and therefore $(*)$ can be further bounded by:

$$(*) \leq \frac{3C_3 VB^2}{2\lambda m} + 2d\exp\left(-C_2(m-1)\right) \tag{26}$$

Plugging the bound in Equation (26) into Equation (25), we obtain:

$$\mathbb{E}_{S \sim D^m}[\#\mathrm{pay}(S)] \leq \frac{3C_3 VB^2}{2\lambda} + 2d\exp\left(-C_2(m-1)\right) \cdot m$$

observe that both terms are bounded by a constant. We thus obtain that there exists a universal constant bounding the expected bidders that pay:

$$\mathbb{E}_{S \sim D^m}[\#\mathrm{pay}(S)] = O\left(\frac{VB^3 M}{\lambda \min\left\{\frac{1}{B}, \frac{c_\ell \cdot \|\boldsymbol{\mu}_D\|}{2(W \cdot K_\ell \cdot B^2 + 2\lambda)}\right\}}\right)$$

Finally, for revenue observe that:

$$\text{revenue}(S) = \sum_{i=1}^{n} p_i \leq V \sum_{i=1}^{n} \mathbb{I}\,[p_i > 0] = V \mathbb{E}_S[\#\text{pay}(S)]$$

and thus $\mathbb{E}_S[\text{revenue}(S)] \leq V \mathbb{E}_S[\#\text{pay}(S)]$. $\qquad\square$

### A.7. $k$-Nearest Neighbors

Let $k \geq 2$ be a positive integer, and let $S = \{(w_i, \boldsymbol{x}_i, y_i)\}_{i=1}^{m}$ be a dataset. A weighted $k$-NN classification algorithm classifies a point $\boldsymbol{x} \in \mathcal{X}$ according to a weighted majority of its $k$ nearest neighbors. Given a dataset $S$ and a point $\boldsymbol{x} \in \mathcal{X}$, we denote by $N_k(\boldsymbol{x}) \subseteq S$ the set of $k$ training points closest to $\boldsymbol{x}$. If the point $\boldsymbol{x}$ appears in the training set $S$, we assume it is included in $N_k(\boldsymbol{x})$, and we also assume that proximity ties are broken in an arbitrarily consistent way. Formally, the weighted $k$-Nearest Neighbors (k$-NN$) classification algorithm makes predictions according to the following rule:

$$h_S(\boldsymbol{x}; S) = \text{sign}\left( \sum_{(w_i, \boldsymbol{x}_i, y_i) \in N_k(\boldsymbol{x})} w_i y_i \right) \tag{27}$$

### A.8. Monotonicity of $k$-NN Accuracy Allocation

**Proposition 3** (Monotonicity of weighted $k$-NN allocation). *For any positive integer $k$, the $k$-NN classification algorithm induces a monotone allocation rule.*

*Proof.* Fix a dataset $S = \{(w_i, \boldsymbol{x}_i, y_i)\}_{i=1}^{m}$, and fix an arbitrary index $i \in [m]$. Denote by $h_S(\boldsymbol{x}_i; b_i)$ the classifier trained on $S$, overriding the weight of coordinate $i$ with $b_i$. The accuracy allocation of point $i$ is therefore $a_i(b_i) = h_S(\boldsymbol{x}_i; b_i) \cdot y_i$. By Equation (27), $h_S(\boldsymbol{x}_i; b_i) \cdot y_i$ is a monotonically non-decreasing function of $b_i$ for any fixed $w_{-i}$, and therefore the allocation rule is monotone as required. $\qquad\square$

**Proposition 4.** *For a dataset $S = \{(w_i, \boldsymbol{x}_i, y_i)\}_{i=1}^{m}$ and a point $i \in [m]$, denote $N^{\backslash i}(\boldsymbol{x}_i)_k = N_k(\boldsymbol{x}_i) \setminus \{(w_i, \boldsymbol{x}_i, y_i)\}$. The incentive-compatible payment of agent $i$ is given by:*

$$p_i(b_i; \boldsymbol{b}_{-i}) = \begin{cases} \sum_{(w, \boldsymbol{x}, y) \in N_k^{\backslash i}(\boldsymbol{x}_i)} wy & \frac{\sum_{(w, \boldsymbol{x}, y) \in N_k^{\backslash i}(\boldsymbol{x}_i)} wy}{w_i y_i} \in (-1, 0) \\ 0 & \text{otherwise} \end{cases}$$

*Proof.* By Myerson's lemma, agent pays their critical bid. Given index $i \in [m]$, denote the weighted sum of agent $i$'s neighbor labels by $f_{S \backslash i}(\boldsymbol{x}_i) = \sum_{(w, \boldsymbol{x}, y) \in N_k^{\backslash i}(\boldsymbol{x}_i)} wy$ . The $k$-NN prediction for agent $i$ is accurate if the total weighted sum, including the agent's own reported weight $w_i$, is identical in sign to the true label $y_i$.

We consider three cases:

- If the agent receives an accurate prediction even when their reported weight is zero ($w_i = 0$), their critical bid is zero. This holds when $f_{S \backslash i}(\boldsymbol{x}_i)$ and $w_i y_i$ have the same sign, formally $f_{S \backslash i}(\boldsymbol{x}_i)/w_i y_i \geq 0$.

- Similarly, if the prediction remains inaccurate even when the agent reports their true valuation, their critical bid is also zero. This holds when $f_{S \backslash i}(\boldsymbol{x}_i)$ is opposite in sign and large in magnitude compared to $w_i y_i$, formally $f_{S \backslash i}(\boldsymbol{x}_i)/w_i y_i \leq -1$.

- Otherwise, the prediction is inaccurate when $w_i = 0$ but becomes accurate at the agent's true valuation $w_i = v_i$. Therefore, there exists a critical weight where the prediction flips from inaccurate to accurate. By definition of the weighted $k$-NN algorithm, this occurs when the weight is exactly equal to $f_{S \backslash i}(\boldsymbol{x}_i)$, which yields the magnitude of critical bid in this case.

Combining the three cases above into a single expression yields the required result. $\qquad\square$

## A.9. $k$-NN Total Payment Lower Bound

*Remark* (Smoothness of valuations distribution). In the proofs below, the term *sufficiently smooth* refers to scalar random variables $W$ having density functions with the smoothness conditions required by Karmarkar et al. (1986). In particular, our analysis holds for density functions with a bounded fourth moment and a cosine transform (real part of the characteristic function) satisfying $\mathbb{E}_{w \sim W}[\cos(wt)] \leq \frac{1}{1+|t|^\gamma}$ for some $\gamma > 0$ (see Karmarkar et al., 1986, Theorem 3.1). Among other distributions, this condition is satisfied by uniform distributions on the unit interval, normal distributions, and density functions with bounded support and sufficiently smooth derivatives. See Karmarkar et al. (1986) for further discussion and extensions.

We recall the classic random partitioning theorem of Karmarkar et al. (1986):

**Theorem 5** (Probabilistic optimal partitioning; Karmarkar et al. (1986, Theorem 3.1)). *Let $W$ be a random variable with bounded support and sufficiently continuous probability density. Given $w_1, \ldots, w_n \sim W$, and some constant $\alpha$ define the optimal partition random variable as: $\Delta_n = \min_{\sigma_1, \ldots, \sigma_n \in \{-1,1\}} |\sum_i \sigma_i w_i - \alpha|$. Then the median value of $\Delta_n$ is $O\left(\frac{\sqrt{n}}{2^n}\right)$.*

Translating to the $k$-NN setting, we show the following:

**Lemma 9.** *Let $W$ be a random variable with bounded support $[0, V]$ and sufficiently continuous density. Denote the median of $W$ by $M > 0$. Given $w_2, \ldots, w_n \sim W$, denote:*

$$y_2, \ldots, y_n = \operatorname*{argmin}_{\sigma_2, \ldots, \sigma_n \in \{-1,1\}} \left| \sum_{i=2}^{n} \sigma_i w_i + \frac{M}{2} \right|$$

*For sufficiently large $k$, it holds that:*

$$\mathbb{P}_{w_1, \ldots, w_n \sim W}\left[ w_1 > \frac{3M}{4} > -\sum_{i=2}^{k} y_i w_i > \frac{M}{4} \right] \geq \frac{1}{4}$$

*Proof.* Denote $\Delta_{k-1} = \sum_{i=2}^{k} y_i w_i$. By Theorem 5, there exist $k_0$ such that for all $k \geq k_0$ the median of $\left|\Delta_{k-1} + \frac{M}{2}\right|$ is smaller than $\frac{M}{4}$. As $w_1$ and $\Delta_{k-1}$ are independent, it holds that:

$$\mathbb{P}_{w_1, \ldots, w_n \sim W}\left[ w_1 \geq M > \frac{3M}{4} > -\Delta_{k-1} > \frac{M}{4} \right] \geq \underbrace{\mathbb{P}_{w_1 \sim W}[w_1 \geq M]}_{=\frac{1}{2}} \cdot \underbrace{\mathbb{P}_{w_2, \ldots, w_n \sim W}\left[ \frac{3M}{4} \geq -\Delta_{k-1} \geq \frac{M}{4} \right]}_{\geq \frac{1}{2}} \geq \frac{1}{4}$$

$\square$

Our revenue lower bounds assume there exist a finite-volume region of the data distribution with some level of label noise and sufficiently smooth distribution of weights:

**Definition 7** (Region with label noise). Let $D \in \Delta(\mathcal{Z})$ be a data distribution. $D$ has a *region with label noise* with parameters $(\varepsilon, \delta, \eta)$ if there exist a point $\boldsymbol{x}_0 \in \mathcal{X}$, constants $\varepsilon, \delta > 0$, and a constant $\eta \in (0, 0.5)$ such that:

1. $\mathbb{P}_{\boldsymbol{x}}[\|\boldsymbol{x} - \boldsymbol{x}_0\| \leq \varepsilon] \geq \delta$

2. $\mathbb{P}_{(w, \boldsymbol{x}, y)}[y = 1 \mid w, \|\boldsymbol{x} - \boldsymbol{x}_0\| \leq 2\varepsilon] \in (\eta, 1 - \eta)$

**Theorem 6** (Revenue lower bound for $k$-NN under label noise, for sufficiently large $k$). *Let $D$ be a data distribution that has a region with label noise with parameters $(\varepsilon, \delta, \eta)$, and assume that for any $\boldsymbol{x}$ such that $\|\boldsymbol{x} - \boldsymbol{x}_0\| \leq \varepsilon$, the distribution of neighbor valuations is sufficiently smooth (see Remark A.9). Then for a sufficiently large fixed $k$ and weighted $k$-NN classification, expected revenue and number of payers grow asymptotically linearly with the number of participants:*

$$\mathbb{E}_{S \sim D^m}[\#\mathrm{pay}(S)] = \Omega(m)$$
$$\mathbb{E}_{S \sim D^m}[\mathrm{revenue}(S)] = \Omega(m)$$

*Proof.* We focus on points $\varepsilon$-close to $\boldsymbol{x}_0$. For the amount of payers, it holds that:

$$
\begin{aligned}
\#\mathrm{pay}(S) &= \sum_{i=1}^{m} \mathbb{I}\left[p_i > 0\right] \\
&\geq \sum_{i=1}^{m} \mathbb{I}\left[\|\boldsymbol{x}_i - \boldsymbol{x}_0\| \leq \varepsilon\right] \cdot \mathbb{I}\left[p_i > 0\right] \\
&\geq \sum_{i=1}^{m} \mathbb{I}\left[\|\boldsymbol{x}_i - \boldsymbol{x}_0\| \leq \varepsilon\right] \cdot \mathbb{I}\left[\forall (w, \boldsymbol{x}, y) \in N_k(\boldsymbol{x}_i) : \|\boldsymbol{x} - \boldsymbol{x}_0\| \leq 2\varepsilon\right] \cdot \mathbb{I}\left[p_i > 0\right]
\end{aligned}
$$

Denote an arbitrary weight-feature-label triplet in the training set by $(w_i, \boldsymbol{x}_i, y_i) \sim D$, and denote by $M > 0$ the median of the neighbor weights distribution. For the expected number of payers $\mathbb{E}_S[\#\mathrm{pay}(S)]$, we will show that agents must pay when certain events hold, and bound the corresponding probabilities from below. Define the following events:

1. Event $A_1$: the random point $\boldsymbol{x}_i$ is $\varepsilon$-close to $\boldsymbol{x}_0$: $\|\boldsymbol{x}_i - \boldsymbol{x}_0\| \leq \varepsilon$

2. Event $A_2$: the $k$ neighbors of $\boldsymbol{x}_i$ are $2\varepsilon$-close to $\boldsymbol{x}_0$: $\forall (w', \boldsymbol{x}', y') \in N_k(\boldsymbol{x}_i) : \|\boldsymbol{x} - \boldsymbol{x}_0\| \leq 2\varepsilon$

3. Event $A_3$: Random neighbor labels satisfy: $y'_1, \ldots y'_{k-1} = \mathrm{argmin}_{\sigma \in \{-1,1\}} \left|\sum_{j=1}^{k} \sigma_j w'_j + \frac{M}{2}\right|$

4. Event $A_4$: It holds that $w_i \geq -\sum_{j=1}^{k} y'_j w'_j \geq \frac{M}{4}$

Taking the expectation over $S \sim D^m$ and applying the definitions above, we obtain:

$$
\begin{aligned}
\mathbb{E}_{S \sim D^m}[\#\mathrm{pay}(S)] &\geq m \cdot \mathbb{P}[A_1 \cap A_2 \cap A_3 \cap A_4] \\
&= m \cdot \mathbb{P}[A_1] \cdot \mathbb{P}[A_2 \mid A_1] \cdot \mathbb{P}[A_3 \mid A_1 \cap A_2] \cdot \mathbb{P}[A_4 \mid A_1 \cap A_2 \cap A_3]
\end{aligned}
$$

We note that the following holds:

1. $\mathbb{P}[A_1]$ is bounded from below by $\delta$ by assumption 1 of the theorem.

2. $\mathbb{P}[A_2 \mid A_1]$ is bounded from below by $\delta^{k-1}$ by assumption 1, as the existence $k-1$ points that are $\varepsilon$-close to $\boldsymbol{x}_0$ ensures that all neighbors of $\boldsymbol{x}_i$ are $2\varepsilon$-close to $\boldsymbol{x}_0$.

3. $\mathbb{P}[A_3 \mid A_1 \cap A_2]$ is bounded from below by $\eta^{k-1}$ by assumption 2 of the theorem.

4. $\mathbb{P}[A_4 \mid A_1 \cap A_2 \cap A_3]$ is bounded from below by $\frac{1}{4}$ for a sufficiently large fixed $k$ by Lemma 9 and assumption 3 of the theorem.

Combining the above, we obtain:

$$
\mathbb{E}_{S \sim D^m}[\#\mathrm{pay}(S)] \geq m \cdot \frac{\delta^k \eta^{k-1}}{4} = \Omega(m)
$$

For revenue, observe that event $A_4$ also guarantees $\sum_{j=1}^{k} y'_j w'_j \leq -\frac{M}{4}$, and therefore agent $i$ needs to bid at least $\frac{M}{4}$ in order to change the sign of the prediction, and from Myerson's lemma we obtain:

$$
\mathbb{E}_{S \sim D^m}[\mathrm{revenue}(S)] \geq m \cdot \frac{\delta^k \eta^{k-1}}{4} \cdot \frac{M}{4} = \Omega(m)
$$

as required. $\qquad\square$

# B. Additional Algorithms

For completeness we give below pseudocode for computing individual payments exactly (Alg. 2) and randomly (Alg. 3), and for the one-shot randomized algorithm (Alg. 4) due to Babaioff et al. (2015).

---

**Algorithm 2** `ComputePaymentExact`

---

    **Input:** User index $i$, sample set $S$, bids $\boldsymbol{b}$, search steps $k$
1: solve $\hat{h}_0 = \arg\min_{h \in H} \mathcal{O}(h; (0, \boldsymbol{b}_{-i}))$
2: **if** $a_i(\hat{h}_0) = 1$ **then**
3:     $p_i \leftarrow 0$
4: **else**
5:     $\underline{z} = 0, \bar{z} = b_i$                                                     ▷ *binary search*
6:     **for** $t = 1, \ldots, k$ **do**
7:         $z_t \leftarrow (\underline{z} + \bar{z})/2$
8:         $\hat{h}_t = \arg\min_{h \in H} \mathcal{O}(h; (z_t, \boldsymbol{b}_{-i}))$
9:         **if** $a_i(\hat{h}_t) = 0$ **then** $\underline{z} \leftarrow z_t$ **else** $\bar{z} \leftarrow z_t$
10:    $p_i \leftarrow \underline{z}$                                               ▷ *highest bid giving $a_i = 0$*
11: **return** $p_i$

---

---

**Algorithm 3** `ComputePaymentRandomized`

---

    **Input:** User index $i$, bids $\boldsymbol{b}$, sample set $S$, number of trials $N$
1: solve $\hat{h}_0 = \arg\min_{h \in H} \mathcal{O}(h; (0, \boldsymbol{b}_{-i}))$                            ▷ *optional step*
2: **if** $a_i(\hat{h}_0) = 1$ **then**
3:     **return** $p_i = 0$
4: $p_i \leftarrow 0$
5: **for** $t = 1, \ldots, N$ **do**
6:     $u \sim \text{Uniform}[0, b_i]$
7:     solve $\hat{h}_u = \arg\min_{h \in H} \mathcal{O}(h; (u, \boldsymbol{b}_{-i}))$
8:     **if** $a_i(\hat{h}_u) = 0$ **then** $p_i \leftarrow p_i + b_i/N$
9: **return** $p_i$

---

---

**Algorithm 4** `ComputeAllPaymentsOneShot` `(Babaioff et al., 2015)`

---

    **Input:** Resampling probability $\mu \in (0, 1)$, bids $\boldsymbol{b}$
1: **for** $i = 1, \ldots, m$ **independently do**
2:     Sample $\gamma_i \sim \text{Uniform}[0, 1]$
3:     Sample $\theta_i \sim \text{Uniform}[0, 1]$
4:     **if** $\theta_i < 1 - \mu$ **then**
5:         $\chi_i \leftarrow 1$
6:     **else**
7:         $\chi_i \leftarrow \gamma_i^{1/(1-\mu)}$
8: $\boldsymbol{b}' \leftarrow (b'_1, \ldots, b'_n)$ where $b'_i = \chi_i b_i$
9: solve $\hat{h}' = \arg\min_{h \in H} \mathcal{O}(h; \boldsymbol{b}')$
10: **for** $i = 1, \ldots, m$ **do**
11:     **if** $\chi_i = 1$ **then**
12:         $p_i \leftarrow b_i \cdot a_i(\hat{h}')$
13:     **else if** $\chi_i < 1$ **then**
14:         $p_i \leftarrow b_i \cdot a_i(\hat{h}') \cdot \left(1 - \frac{1}{\mu}\right)$

---

### B.1. Discarding Examples in Payment Computations With SVM

*Proof of Lemma 3.* Denote by $\beta$ the classification stability of standard (unweighted) SVM and $\beta_i$ to be the classification stability of weighted SVM when leaving out data point $i$. Consider a user $i \in [m]$, and assume w.l.o.g. that $y_i = 1$. We use the convention that the margins are of unit size (hence the 1 in the condition). First, we'll prove that if for any $j$ it holds that $f(x_j; \boldsymbol{b}) > 1 + 2\beta_i$, then for all $z \in [0, b_i]$ this results in $f(x_j; (z, \boldsymbol{b}_{-i})) > 1$, i.e., $x_j$ remains outside the margin.

Let $z \in [0, b_i]$:

$$
\begin{aligned}
\left| f(x_j; \boldsymbol{b}) - f(x_j; (z, \boldsymbol{b}_{-i})) \right| &= \left| f(x_j; \boldsymbol{b}) - f(x_j; (0, \boldsymbol{b}_{-i})) + f(x_j; (0, \boldsymbol{b}_{-i})) - f(x_j; (z, \boldsymbol{b}_{-i})) \right| \\
&\leq \left| f(x_j; \boldsymbol{b}) - f(x_j; (0, \boldsymbol{b}_{-i})) \right| + \left| f(x_j; (0, \boldsymbol{b}_{-i})) - f(x_j; (z, \boldsymbol{b}_{-i})) \right| \\
&\underset{(*)}{\leq} \beta_i(\boldsymbol{b}) + \beta_i((z, \boldsymbol{b}_{-i})) \underset{(**)}{\leq} 2\beta_i(\boldsymbol{b}).
\end{aligned}
$$

Where the inequality $(*)$ is by the definition of weighted classification stability, and $(**)$ holds since $\forall z \in [0, b_i] : \beta_i(\boldsymbol{b}) = \beta b_i \geq \beta z = \beta_i((z, \boldsymbol{b}_{-i}))$. From the above we can derive:

$$
f(x_j; (z, \boldsymbol{b}_{-i})) - f(x_j; \boldsymbol{b}) \geq -2\beta_i
$$

Since $f(x_j; \boldsymbol{b}) > 1 + 2\beta_i$, we get $f(x_j; (z, \boldsymbol{b}_{-i})) > 1$ as required.

Now, we'll show that the solution of both $\mathcal{O}_{\text{SVM}}((z, \boldsymbol{b}_{-i}))$ and $\mathcal{O}_{\text{SVM}}((z, \boldsymbol{b}_{-i}^{(i)}))$ is identical for all $z \in [0, b_i]$. This will entail equivalent allocations and hence equivalent payments.

Notice that for all $j \in [m]$ with $f(x_j; (z, \boldsymbol{b}_{-i})) > 1$, the correctness constraint in $\mathcal{O}_{\text{SVM}}((z, \boldsymbol{b}_{-i}))$ is tight, i.e., the corresponding slack variable $\xi_j$ is 0. Therefore, point $j$ does not contribute to the loss term in the objective. Hence, changing the weight of point $j$ does not change the solution. As a result, solving $\mathcal{O}_{\text{SVM}}((z, \boldsymbol{b}_{-i}^{(i)}))$ will give the same solution.

$\square$

### B.2. Non-Negativity of Payments in Algorithm 3

In the original Alg. 3, as it appears in (Archer et al., 2004), the payment is set to be $b_i - ZX$ where $Z = b_i * 1[a_i(u, b_{-i}) = 1] \in \{0, b_i\}$ meaning it is set according to the result of the auction run on bids $(u, b_{-i})$ and $X$ is an independent unbiased estimator for the inverse of the overall probability that agent $i$ wins their desired allocation. This payment rule can produce a negative payoff, specifically when $X \in (0, 1)$. However, since our allocations are deterministic, $ZX$ are reduced to a an indicator, $ZX = b_i * 1[a_i(u, b_{-i}) = 1]$, therefore negative payoffs are not possible in our setting.

## C. Experimental Details

All code and experiments are implemented in Python. Code is attached as supplementary material, and also provided at `https://github.com/BML-Technion/accuracy_auctions`. All results are averaged over multiple random data generations or splits. The number of trials varied from 5-100, depending on the experimental setting and need.

### C.1. Learning Algorithms

Our experiments used linear SVM (synthetic and real data) and logistic regression with $\ell_2$ regularization (real data). For implementation we used sklearn for both, which supports example weighing. In our synthetic experiments the regularization coefficient $\lambda$ (or $C = 1/\lambda$ in the implementation) was set to the default value of 1, though this had little effect on results. In our semi-synthetic experiments $\lambda$ was fine-tuned to prevent overfitting.

Specifically, Sec. 7.1 and Appendices D.2, D.5 use linear SVM with hinge loss. Sec.7.2 use logistic regression with $L_2$ regularization. Appendix. D.4 uses linear SVM with both hinge and squared hinge loss in addition to logistic regression with $L_2$ regularization . Appendix D.6 also uses KNN implemented using sklearn with different values of $k$.

### C.2. Real Data

Our semi-synthetic experiments are base on the `folktables` dataset (Ding et al., 2021). This data is publicly available at: `https://github.com/socialfoundations/folktables`. Specifically, we used the Income task where the goal is

to predict wether an individual's income exceeds $50K$. We focused our analysis on New Jersey (NJ) since, in this state, the distribution of the target variable (PINCP > \$50K) was approximately balanced. This choice minimizes issues related to class imbalance and allows us to construct larger balanced training datasets without excessive subsampling. The data includes $\sim 44,000$ examples; of these, in each trail we use a random subset of $30,000$ examples split 60-40 into train and validation sets.

**Features.** We used the following standard features: COW (class of worker), SCHL (educational attainment), MAR (marital status), OCCP (occupation code), RELP (relationship), SEX, RAC1P (race), AGEP (age), DIS (disability status), ESR (employment status), HISP (Hispanic origin), WKHP (usual hours worked per week), and MIG (migration status).

**Distribution of values.** We experimented with two different ways to set individual values $v$. Both are parameterized by a "global" value $v_+$, which allows to control variation in values.

- **Feature-based:** Here we set $v = v(z)$ where $z$ is the WKHP attribute of the dataset. Note $z$ was not included in features $x$. Since a typical work-week is 40 hours, we set $v = 1$ if $z < 40$, $v = 2$ if $z = 40$, and $z = v_+$ if $z > 40$. The corresponding ratios of values are roughly 43%, 31%, and 26%, respectively.

- **Label-based:** Here we set $v = v(y)$ as $v(0) = 1$ and $v(1) = v_+$.

**Preprocessing.** Data was imported using the `folktables` API. Ordinal variables were retained without modification, while selected demographic variables were converted into binary indicators using rule-based mappings. The occupation feature was coarsened into 14 broad occupational groups and subsequently one-hot encoded. To obtain a smoother representation simulate a continuous distribution, small normal noise ($\mu = 0, \sigma = 0.4$) was added to categorical and 1-hot entries.

## D. Additional Experimental Results

### D.1. Welfare Accuracy Gap – Constructive Example

To understand the gap between accuracy and welfare, we give a simple illustrative example. Consider a one-dimensional classification problem over threshold classifiers, $h(x;\theta) = \mathbb{1}\{x < \theta\}$. Points $x$ are sampled uniformly from $[0, 1.5]$, and labels $y$ are defined as:

$$y = 0 \text{ if } x \in [0, 0.5], \qquad y \sim \text{Ber}(0.5) \text{ if } x \in [0.5, 1], \qquad y = 1 \text{ if } x \in [1, 1.5]$$

Figure 4a depicts a sampled dataset as described above. For simplicity, assume values correlate with labels as $v_y$ with $v_0 = 1$ and $v_1 = 3$. An incorrect prediction entails $v_y = 0$ for both classes. Fig. 4b shows accuracy and welfare for all possible $\theta$. In terms of accuracy, any threshold $\theta \in [0.5, 1]$ is optimal since it attains the Bayes optimal risk. However, since values relate to labels, welfare breaks this symmetry, and the welfare-optimal classifier is at $\theta = 1$.Notice that here the welfare-optimal classifier is also accuracy-optimal (but not vice versa). This exemplifies problems in which maximizing accuracy could result in lower welfare.

### D.2. Payment Algorithms – Comparison Between Exact and Randomized

Table 2 compares payments computed under the different algorithms proposed in Sec. 6. Here we use the same setup as in our first synthetic experiment. The exact algorithm provides correct payments (up to numerical approximation) and so can serve as a benchmark. Note the number of calls to $\mathcal{O}$ can be less than one because relatively few points satisfy the condition in Lemma 2 and require explicit payment computation. The random algorithm gives payments that are close in expectation, and similar welfare. Increasing $N$ provides a tighter approximation, with only a slight increase in runtime. Notice the speedup relative to the exact baseline is only mild. The oneshot baseline computes payments at a fraction of runtime; this, however, comes at a steep increase in both bias and variance in payments, and leads to some extreme payments—both positive and negative (i.e., rebates). Both bias and variance can be reduced by increasing $\mu$, but this comes on account of a (small) reduction in welfare.

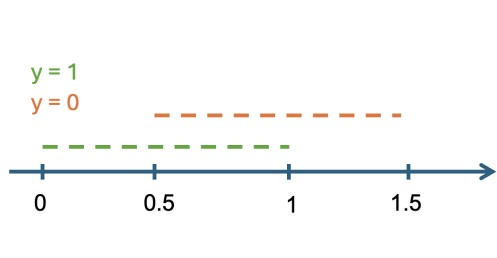

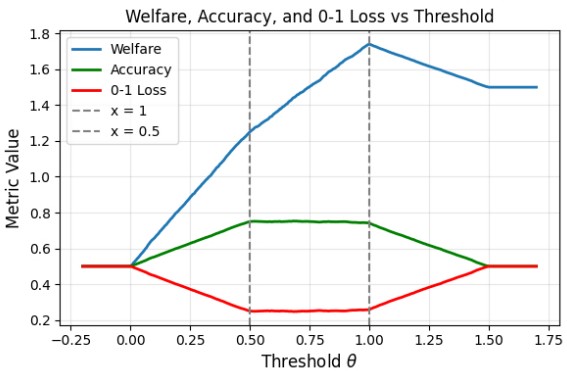

*(a)* The sampled data

*(b)* Accuracy and average welfare as a function of $\theta$

*Figure 4.* 1D example with threshold classifiers.

| payment algorithm | total payments ($m = 1000$) mean | stdev | min | max | mean calls to $\mathcal{O}$ | mean welfare |
|---|---|---|---|---|---|---|
| exact | 0.003799 | 0.076 | 0.137 | 2.147 | 0.203 | 0.785 |
| random $N = 1$ | 0.003899 | 0.085 | 1.000 | 3.000 | 0.125 | 0.783 |
| random $N = 3$ | 0.003832 | 0.080 | 0.333 | 3.000 | 0.133 | 0.783 |
| random $N = 6$ | 0.003813 | 0.078 | 0.167 | 3.000 | 0.145 | 0.783 |
| random $N = 9$ | 0.003820 | 0.077 | 0.111 | 3.000 | 0.157 | 0.783 |
| oneshot $\mu = 0.25$ | 0.010273 | 2.081 | 1.000 | 3.000 | 0.001 | 0.781 |
| oneshot $\mu = 0.4$ | -0.001515 | 2.079 | 1.000 | 3.000 | 0.001 | 0.780 |
| oneshot $\mu = 0.5$ | -0.002303 | 2.081 | 1.000 | 3.000 | 0.001 | 0.781 |
| oneshot $\mu = 0.6$ | -0.000384 | 2.080 | 1.000 | 3.000 | 0.001 | 0.780 |
| oneshot $\mu = 0.75$ | -0.003162 | 2.081 | 1.000 | 3.000 | 0.001 | 0.781 |

*Table 2.* Payment algorithms – comparison between exact and randomized

### D.3. Payment Algorithms – Theoretical Comparison Between Exact and Randomized

The following table compares the number of calls to $\mathcal{O}$ for each of the payment algorithms proposed in Section 6. We define $m$ as the number of agents, $r$ as the number of relevant agents, $z$ as the number of paying agents i.e. number of relevant agents with $p > 0$, $k$ as the number of search iterations in binary search, and $n$ as the number of rounds of randomness. Notice that in case $k$ is chosen to be relatively small (e.g. $k = 33$ in Algorithm 2), the difference between the number of calls to $\mathcal{O}$ in the exact case versus the random case will not be substantial.

| Payment algorithm | # calls to $\mathcal{O}$ |
|---|---|
| exact | $1 + r + z \cdot k$ |
| random | $1 + r + z \cdot n$ |
| oneshot | $1$ |

### D.4. Varying Loss and $\lambda$ with Increasing $m$

Using real data (see C.2), we study the interaction between loss functions, the regularization parameter $C$, the sample size $m$, and their combined effect on the performance of weighted SVMs and the resulting number of payers. the table below summarizes results across three loss functions, hinge, log, and squared hinge, over multiple scaling regimes for $C = c(m)$ and increasing values of $m$, reporting the number of payers, their percentage, and test accuracy.

Overall, the results, summarized in Table 3, indicate that log loss and squared hinge loss produce relatively similar behavior

across increasing m, with only minor fluctuations in both the number and proportion of payers. In contrast, hinge loss yields a substantially higher number of payers across most configurations, although this effect is sensitive to the scaling of $C$. In particular, for certain choices of $c(m)$, the proportion of payers under hinge loss remains nearly invariant as $m$ increases.

| loss | c(m) | Number of Payers | | | | Percent Pay | | | | Test Accuracy | | | |
|---|---|---|---|---|---|---|---|---|---|---|---|---|---|
| | | 1000 | 2000 | 4000 | 10000 | 1000 | 2000 | 4000 | 10000 | 1000 | 2000 | 4000 | 10000 |
| hinge | $m$ | 215.700 | 408.100 | 837.100 | 2057.900 | 21.570 | 20.405 | 20.928 | 20.579 | 0.629 | 0.618 | 0.625 | 0.600 |
| | $m/10^2$ | 66.500 | 267.900 | 678.800 | 1981.600 | 6.650 | 13.395 | 16.970 | 19.816 | 0.699 | 0.665 | 0.661 | 0.612 |
| | $m/10^4$ | 16.100 | 14.200 | 21.800 | 73.200 | 1.610 | 0.710 | 0.545 | 0.732 | 0.692 | 0.706 | 0.716 | 0.714 |
| | $m/10^5$ | 12.200 | 13.400 | 19.400 | 20.300 | 1.220 | 0.670 | 0.485 | 0.203 | 0.680 | 0.697 | 0.709 | 0.714 |
| | $\sqrt{m}/10$ | 27.400 | 62.500 | 140.000 | 665.900 | 2.740 | 3.125 | 3.500 | 6.659 | 0.705 | 0.714 | 0.718 | 0.705 |
| | $\sqrt{m}/10^2$ | 22.400 | 19.900 | 31.500 | 73.200 | 2.240 | 0.995 | 0.788 | 0.732 | 0.700 | 0.710 | 0.715 | 0.714 |
| | $\sqrt{m}/10^3$ | 14.300 | 17.700 | 19.200 | 20.300 | 1.430 | 0.885 | 0.480 | 0.203 | 0.686 | 0.702 | 0.712 | 0.714 |
| | $\sqrt{m}/10^5$ | 2.600 | 5.800 | 9.100 | 13.400 | 0.260 | 0.290 | 0.228 | 0.134 | 0.598 | 0.662 | 0.679 | 0.691 |
| log | $m$ | 16.700 | 16.500 | 19.200 | 18.900 | 1.670 | 0.825 | 0.480 | 0.189 | 0.706 | 0.712 | 0.717 | 0.718 |
| | $m/10^2$ | 16.900 | 16.500 | 18.700 | 18.400 | 1.690 | 0.825 | 0.468 | 0.184 | 0.706 | 0.712 | 0.716 | 0.718 |
| | $m/10^4$ | 12.400 | 15.300 | 17.400 | 20.200 | 1.240 | 0.765 | 0.435 | 0.202 | 0.695 | 0.707 | 0.714 | 0.718 |
| | $m/10^5$ | 9.800 | 12.900 | 15.800 | 17.300 | 0.980 | 0.645 | 0.395 | 0.173 | 0.684 | 0.698 | 0.708 | 0.716 |
| | $\sqrt{m}/10$ | 17.300 | 16.100 | 17.900 | 18.300 | 1.730 | 0.805 | 0.447 | 0.183 | 0.705 | 0.712 | 0.717 | 0.718 |
| | $\sqrt{m}/10^2$ | 15.100 | 17.600 | 17.900 | 20.200 | 1.510 | 0.880 | 0.447 | 0.202 | 0.701 | 0.710 | 0.716 | 0.718 |
| | $\sqrt{m}/10^3$ | 11.500 | 15.700 | 15.900 | 17.300 | 1.150 | 0.785 | 0.398 | 0.173 | 0.690 | 0.702 | 0.711 | 0.716 |
| | $\sqrt{m}/10^5$ | 2.800 | 3.500 | 5.900 | 9.500 | 0.280 | 0.175 | 0.148 | 0.095 | 0.616 | 0.658 | 0.678 | 0.691 |
| hinge$^2$ | $m$ | 19.000 | 16.900 | 14.000 | 16.000 | 1.900 | 0.845 | 0.350 | 0.160 | 0.704 | 0.710 | 0.715 | 0.716 |
| | $m/10^2$ | 18.900 | 16.600 | 14.500 | 17.300 | 1.890 | 0.830 | 0.362 | 0.173 | 0.704 | 0.710 | 0.715 | 0.716 |
| | $m/10^4$ | 17.800 | 17.200 | 14.800 | 16.700 | 1.780 | 0.860 | 0.370 | 0.167 | 0.704 | 0.709 | 0.714 | 0.716 |
| | $m/10^5$ | 12.600 | 16.200 | 17.100 | 17.300 | 1.260 | 0.810 | 0.428 | 0.173 | 0.695 | 0.704 | 0.713 | 0.715 |
| | $\sqrt{m}/10$ | 19.200 | 16.800 | 14.600 | 16.300 | 1.920 | 0.840 | 0.365 | 0.163 | 0.704 | 0.710 | 0.715 | 0.716 |
| | $\sqrt{m}/10^2$ | 19.400 | 16.500 | 14.300 | 16.700 | 1.940 | 0.825 | 0.358 | 0.167 | 0.704 | 0.710 | 0.715 | 0.716 |
| | $\sqrt{m}/10^3$ | 16.200 | 17.000 | 16.900 | 17.300 | 1.620 | 0.850 | 0.422 | 0.173 | 0.699 | 0.708 | 0.714 | 0.715 |
| | $\sqrt{m}/10^5$ | 5.600 | 8.300 | 13.400 | 14.100 | 0.560 | 0.415 | 0.335 | 0.141 | 0.671 | 0.689 | 0.694 | 0.701 |

*Table 3.* Varying loss and $\lambda$-regularization parameter with increasing $m$- dataset sizes.

## D.5. Bias-Variance of Values

We conduct a simulation to examine whether heterogeneity in agent valuations—specifically, the variance of the agent weights—affects average payments and percent of payers. e use multivariate Gaussian class-conditional distributions $P(x|y) = \mathcal{N}(\boldsymbol{\mu}_y, \sigma^2 I_d)$ with means $\boldsymbol{\mu}_y$ and isotropic covariance parameterized by $\sigma$, and set $P(y=1) = P(y=0) = 1/2$. We use $d=8$, fix $\|\boldsymbol{\mu}_1 - \boldsymbol{\mu}_0\| = 1$, the mean of the values $v_i$ is set to 1.

The payments are calculated with a constant case ($v_i = 1$ for all $i$) and gamma-distributed cases with varying shape parameters $k \in \{16, 4, 1, 0.5, 0.33\}$, which induce increasing variance in valuations while holding the mean fixed at one. For each configuration, we run weighted SVM with hinge-loss. The results, shown in Figure 5, indicate that higher valuation variance (i.e., lower k) leads to a slight increase in average payments, while the proportion of paying agents remains effectively unchanged across both variance levels and dataset sizes.

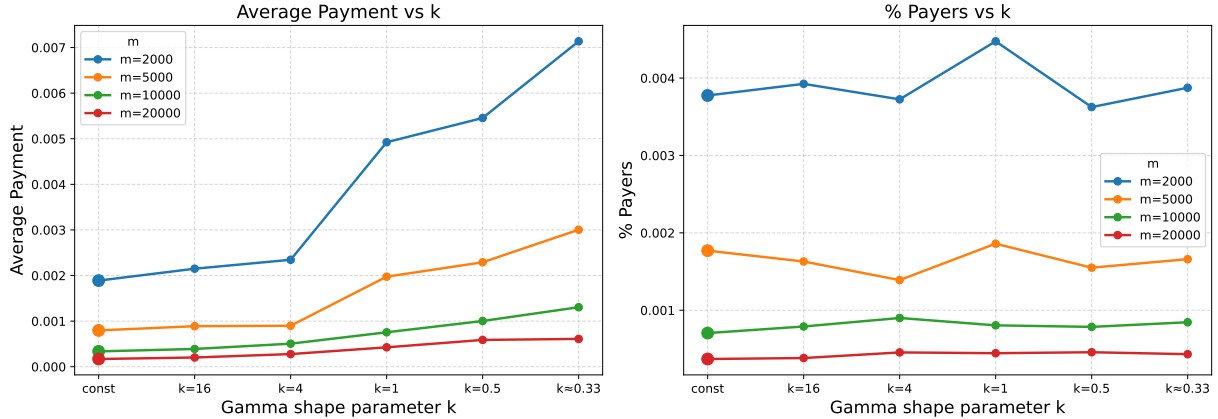

*Figure 5.* Payments and % Payers under varying valuation variance and dataset sizes $m$.

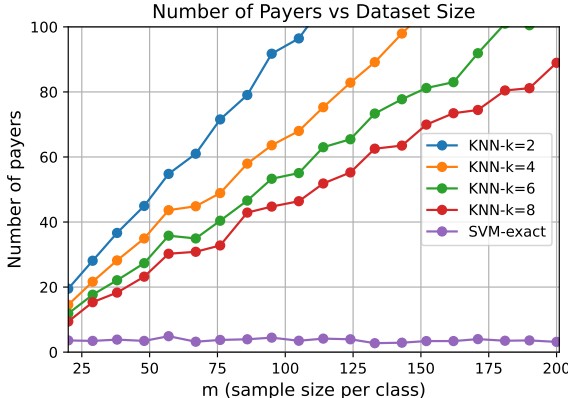

*Figure 6.* Comparing number of payers in SVM vs. $k$-NN (Appendix D.6).

### D.6. Number of Paying Agents: SVM vs. $k$-NN

In Section 5, we derived an upper and lower bound on the number of payers by considering weighted SVM and weighted $k$-NN. Here we provide empirical support to our theoretical results. We use multivariate Gaussian class-conditional distributions $P(x|y) = \mathcal{N}(\boldsymbol{\mu}_y, \sigma^2 I_d)$ with means $\boldsymbol{\mu}_y$ and isotropic covariance parameterized by $\sigma$, and set $P(y = 1) = P(y = 0) = 1/2$. We use $d = 2$, fix $\|\boldsymbol{\mu}_1 - \boldsymbol{\mu}_0\| = 0.5$, and for simplicity set $v_i = 1$ for all $i$. Gaussian distributions have full support and therefore overlap, and thus the data distribution has regions with label noise.

Results are presented in Figure 6. We observe an approximately linear increase in the number of payers when using $k$-NN, while using SVM shows a bounded number of payers, in alignment with our theoretical findings.

