# OpenReview forum: "Welfare-Optimal Classification with Accuracy Auctions"
_ICML.cc/2026/Conference — ICML 2026 spotlight_

### Official Review · Reviewer_FzyY · 2026-03-04

**Soundness:** 3
**Presentation:** 2
**Significance:** 2
**Originality:** 3
**Overall Recommendation:** 3
**Confidence:** 4

**Summary:**

This paper studies the problem of resource allocation in machine learning predictions, arguing that the standard objective of maximizing expected accuracy fails to account for heterogeneous user preferences. As a result, the authors propose optimizing for social welfare, defined as the average individualized gain users derive from correct predictions. Because users' valuations are private and subject to strategic overreporting, the authors formulate this task as an "accuracy auction". This study can be categorized into three parts: Mechanism Design, Theoretical Bounds, and Algorithmic & Empirical Validation. In the first part, the authors introduce the core mechanism such as dominant-strategy incentive-compatible mechanism. In the second part, the authors show relationship between sample size and expected number of paying users, for example, constant for SVM. In the last part, the authors propose 3 practical algorithms to compute payments. Experiments on synthetic data validate the theoretical revenue bounds, and evaluations on the Folktables dataset demonstrate the practical tradeoffs between accuracy and social welfare.

**Compliance With Llm Reviewing Policy:**

Affirmed.

**Ethical Review Concerns:**

I am flagging this paper for ethics review due to structural fairness and discrimination concerns.
The paper implicitly equates a user's monetary valuation ($v_i$) with their true underlying utility, operating under the standard assumption of quasi-linear utility without budget constraints. However, in reality, a user's bid is strictly bounded by their monetary constraints. A user might possess a high intrinsic need for a correct prediction but have an artificially low $v_i$ due to an inability to pay. If such a mechanism were deployed in critical tasks (e.g., medical diagnostics or AI-assisted clinical documentation), it would systematically bias the model's accuracy toward wealthier users while degrading accuracy for low-income demographics, framing this wealth-weighted allocation as "maximizing social welfare."

**Ethical Review Flag:**

Flag this paper for an ethics review.

**Ethics Expertise Needed:**

["Discrimination / Bias / Fairness Concerns"]

**Key Questions For Authors:**

1. In section 6, the authors claim that the randomized payment algorithm introduces "the possibility of instances with negative payoff," which implies that the mechanism satisfies ex-ante ir (in expectation) and violates ex-post ir. However, following Algorithm 3, if a user bids trustfully ($b_i$ = $v_i$) and $1 \leq N$, the maximum penalty is bounded at $b_i$. Thus, the worst-case ex-post payoff is 0. Can you provide an example on negative payoff under Algorithm 3 to clarify the “contradiction” of algorithm’s theoretical guarantees?

2. In the Allocation section, "Fig. 4 compares accuracy and welfare for users from each class and overall..." seems incorrect. The interpretation seems to align better with Fig 2 (right). Can you provide explanation on how your interpretation matches with Fig 4?

3. The Impact Statement acknowledges that real-world users often behave irrationally, a reality that fundamentally challenges many classical economic theories. As a thought experiment, consider a realistic scenario in a high-stakes deployment (e.g., medical or financial predictions). In such cases, a user with a true valuation $v_i = 100$ might panic due to loss aversion and heuristically overbid to an extreme value, such as $b_i = 10,000$. According to your derivations (e.g., Lemma 7), the classification stability margin $\beta$ is proportional to the maximum weight $w_{max}$. If irrational overbidding causes an explosion in $w_{max}$, the stability boundary $\beta$ would significantly widen, capturing far more users. How robust is the mechanism's welfare optimality and its constant-bound guarantee when the strict assumption of perfect user rationality is relaxed even a small fraction of users overbid? To draw a parallel, the classical shift from Expected Utility theory to Prospect Theory demonstrates how relaxing rationality assumptions can lead to drastically different conclusions, especially when scenarios under Prospect Theory are more common in reality.

**Limitations:**

yes

**Strengths And Weaknesses:**

Strengths:
The paper introduces a original perspective on classification accuracy as a limited, contested resource, which should be allocated to maximize social welfare rather than just accuracy. The connection between algorithmic stability and Myerson auction revenue is also a solid contribution to the intersection of mechanism design and machine learning algorithms. The proofs on the expected number of paying users for linear SVM and k-NN are rigorous with sufficient intuition for understanding. The set up of practical algorithms is also sound, especially on how Lemma2 can efficiently speed up searching for candidate paying users.

Weakness:
There are several typos in writing, for example, misspelling Myerson’s lemma in Equation 6. Furthermore, several notation formulation and interpretations can be more clear for better understanding, e.g., Equation 6. Derivative for a discontinuous step function a(.) is mathematically imprecise in standard calculus. Rewriting using a Riemann-Stieltjes integral or explicitly through Dirac delta function can be considered. Reasoning on “the possibility of instances with negative payoff (though individual rationality still holds)” is not straightforward. Claims on “Thm. 2 states that the number of paying users is bounded by a constant” is not straightforward, and better reasoning can starting from explicitly mention both the user count and revenue bounds are constant bounded to avoid confusion. In Section 6, the phrase "finding the threshold of $a_i$" is mathematically imprecise. A step function itself does not have a threshold in this context; rather, $b_i^c$ acts as the threshold for the variable $z$. Could be rewording to “…to finding the threshold value of $z$ at which the allocation $a_i(.)$ transitions from 0 to 1.” The key assumption on rationality needs better justification given the authors claim “While the type of assumptions we make are common in economics, it is well known that in reality, true users are rarely rational. Human-subject experiments have shown that this holds in particular for truthful auctions, and despite users being aware that truthful reporting is by definition the optimal strategy (e.g., Noti et al., 2014).” This significant gap between economic theory and human behavior somewhat limits the practical significance of the proposed system in real-world deployments.
A minor suggestion would be to explicitly stating that SVM and k-NN represent the extreme ends of the uniform stability for a broader ML audience.

---

> ### Author Rebuttal · Authors · 2026-03-31
>
> Thank you for your careful review. As we understand, it includes: (i) suggestions for improving clarity and precision, (ii) concerns regarding the rationality assumption, and (iii) a possible (ethical) monetary interpretation concern. We address each of these in detail below. Although some of these appear to regard mechanism design in general, rather than our paper in particular, we hope our response will help alleviate your concerns.
>
> Weaknesses:
>
> > There are several typos in writing
>
> Thanks for spotting these – we will find and correct them.
>
> > Several notation formulation and interpretations can be more clear
>
> Thanks, we will clarify these and make all statements more precise.
>
> > Claims on “Thm. 2 states that the number of paying users is bounded by a constant” is not straightforward
>
> By constant we mean independent of sample size $m$, which is our parameter of interest. This is mentioned in the abstract, intro, and the beginning of Sec. 5.1.
>
> > The key assumption on rationality needs better justification… This significant gap …  limits the practical significance of the proposed system
>
> If we understand correctly, this regards classical mechanism design in general, and is not particular to our setting. Given the scope and significance of this well-established field, we believe there is value in studying problems in machine learning under its perspective, both theoretically and as a basis for future applied work. We also believe it is beneficial to first establish results for the rational setting, as a basis for future irrational extensions and practical toolsets. We view our work as a first such step for accuracy auctions.
>
>
> > A minor suggestion would be to explicitly stating that SVM and k-NN represent the extreme ends of the uniform stability
>
> Thanks for this suggestion, we will highlight this perspective explicitly.
>
>
> Questions:
>
> > In section 6, the authors claim that the randomized payment algorithm introduces "the possibility of instances with negative payoff”
>
> Thank you for pointing this out. The possibility of negative ex-post utility stems from the original (conservative) analysis of Archer & Tardos (2004). However, upon further inspection, we can show that the particular structure of our setting entails *no negative payoffs* in any case. Specifically, the original algorithm requires two (decoupled) sources of randomness: one for payments, and one for allocations, and negative payments can materialize through a combination of both. Our allocations are deterministic, which implies that payments are always non-negative. We will add the new proof to the Appendix.
>
>
> > In the Allocation section, Fig. 4 … The interpretation seems to align better with Fig 2 (right)
>
> Apologies – this is a typo, and should read Fig. 2 instead of Fig. 4. We will fix this. Thank you for noticing!
>
>
> > The Impact Statement…
>
> This is an interesting question. First, note that if $i$ overbids then this will increase only its own margin $\beta_i$ (Lemma 2), and so the condition on which (truthful) users might pay (as a function of their distance to the boundary) does not change. Lemma 7 uses the global $w_{max}$ for convenience; the bound applies pointwise to each $\beta_i$ via $w_i$. Second, note that for any user $i$ with $y_i f(x_i) > \beta_i$, overbidding has no effect on outcomes at all. Third, for all other points, the effect of overbidding is mediated by stability; that is, it reduces with $m$. Fourth, at least intuitively, if overbidding behavior is symmetric across label classes, then we expect biases to “balance out” and reduce the effect.
>
> More generally, while we agree that accounting for irrational behavior is important, we hope you agree that it is also important to first establish results for the rational case, and then build on this to extend and robustify the mechanism. We believe this applies to any mechanism in general and not only to ours.
>
>
> Ethics:
>
> > The paper implicitly equates a user's monetary valuation with their true underlying utility
>
> While values in our setting can be monetary, they **need not necessarily be such**. Because our framework maximizes welfare (and disregards profit), “payments” should be interpreted as any form of utility reduction, which can be indirect. For example, the system may implement payments by injecting noise into predictions, or introduce friction into corresponding decisions (e.g., longer wait times, lower quality recommendations) when these are quantifiable utility-wise. This is similar to ideas introduced in [1].
>
> Here as well we believe your concern regards mechanism design more generally, and is not specific to our setting or approach. However, we agree that care should be taken in regards to fairness in general—and monetary implications in particular—and will gladly elaborate on this point when discussing values and payments in Sec. 3 and in the impact statement.
>
> [1] Hartline & Roughgarden. Optimal Mechanism Design and Money Burning. STOC 2008.

---

> > ### Author Rebuttal · Reviewer_FzyY · 2026-04-03
> >
> > Thank you for the detailed and respectful rebuttal. My concerns related to Algorithm 3 and minor typos have been sufficiently addressed.
> > However, I remain concerned about the practical significance and ethical implications (Question 3& Ethics), which I think go beyond “general mechanism design problems.”
> > In your rebuttal, you proposed that values need not be monetary citing work by H&R. In reality, particularly in the social science or health domains, implementing “wait times” or "lower quality" exacerbates the fairness issue. A well-known phenomenon in social economy is time poverty. If the proposed alternative to monetary bidding is to intentionally degrade the user experience or delay critical ML predictions, how does this framework safely advance the deployment of machine learning in the real world? Perhaps scoping the claims in the present work away from high-stakes ML domains, for example healthcare, can better justify the contribution to more proper domains?
> > I agree that “it is important to first establish results for the rational case, and then to extend and robustify the mechanism,” but I suggest incorporating this philosophical discussion into the main text by expanding statements in Limitations or Impact Statement.

---

> > > ### Author Response · Authors · 2026-04-07
> > >
> > > Thank you for engaging with us and for clarifying your point on fairness considerations. Overall we will be happy to add a discussion on the scope of applicability of our framework in this regard. We agree this is important and will benefit the paper. We also appreciate your insistence on the need to discuss and raise awareness of these aspects (which, if we understand correctly, is not limited to a possible monetary interpretation, as implied by your initial review).
> > >
> > > Nonetheless—and without reducing our commitment above—we would like to better explain our perspective, as we believe it provides important context and scope for the concern itself.
> > >
> > > As any method applied in a social setting, the decision of whether to use it or not in a certain domain requires careful deliberation. Ideally, a policymaker should weigh the risks and benefits of the approach, compare it to other possible alternatives, and decide on the approach which best serves public interest.
> > >
> > > Our intent was to provide policymakers the information necessary to make such informed decisions: by being maximally transparent about our assumptions, and by accurately portraying both the benefits and limitations of our approach. In terms of limitations, we thank you for your comments as we believe they shed light on an important perspective which we may have not sufficiently stressed. In particular, we will stress the need to determine on a per-domain basis whether payments are suitable, and if so, of which type.
> > >
> > > However, as for benefits, we would like to highlight several aspects which we think are relevant to your concern, and which may have been overlooked:
> > >
> > > **Generalization**. Firstly, all concerns regarding payments apply only to users in the train set, since only these (maybe) pay. Once a classifier has been learned, guarantees on welfare hold indefinitely; and if optimized, then the test-time population enjoys high welfare *for free*. This means that in domains where fairness concerns arise, a policymaker should weigh: (a) the social cost or fairness implications *for the finite set of train-time users*, versus (b) the welfare benefits *of the entire population*.
> > >
> > > **Alternative policies**. Secondly, for policymakers seeking to promote welfare using machine learning, accuracy auctions offer an alternative to an existing prevalent and widespread default approach, which is to maximize accuracy. As we argue, maximizing accuracy—by ignoring the issue of private valuations, and implicitly assuming uniform values—can result in very low expected welfare. The policymaker can now choose between: (a) maximizing population accuracy without payments (and hoping that welfare is reasonable), or (b) maximizing population welfare but paying the social cost of having some train-time users pay (in some form). Note both approaches make *some* assumption on human behavior and agency (for accuracy maximization—that there is none), each entailing its own risks.
> > >
> > > **Auctions and payments**. Our third point is that we still believe your concern applies to auctions in general, and is not specific to auctions where the limited resource is accuracy. For example, we think your point on poverty applies equally to treatment decisions based on predictions, and to treatment decisions made directly (without relying on predictions). An auction that allocates treatment decisions directly should be subject to the same (valid) arguments raised. This is not to reduce the significance of the arguments—only to state they hold generally, and are not specific to prediction-based allocations (in contrast to our two points above, which are).
> > > This is important because Myerson’s lemma implies that payments are *necessary* for maximizing welfare via truthful elicitation when users are rational (again, see H&R for discussion). In this context, and from a policy perspective, we think our results are optimistic: only train time users (possibly) pay; payments need not be monetary; classification-stable algorithms ensure a constant number paying users in theory; and simulated results suggest that empirically often only a handful do. Thus, even in domains where fairness risks exist, their extent is limited.
> > >
> > > Finally, and as we noted, we fully agree there is merit in developing welfare-maximizing approaches that support non-rational behavior. We also find the question of whether social burden can be distributed differently very intriguing and non-trivial. However, we do not see why the importance of these future directions should diminish the potential value of our current work to the learning community.

---

### Official Review · Reviewer_TSuQ · 2026-03-10

**Soundness:** 4
**Presentation:** 3
**Significance:** 3
**Originality:** 3
**Overall Recommendation:** 5
**Confidence:** 4

**Summary:**

This paper studies the problem of binary classification with the novel objective of maximizing welfare, where welfare is based on agents’ values for correct label (rather than a positive label). The agent labels are unknown to themselves and values are private. The setup is that bidders are drawn to create the training sample, and an auction (which is strategyproof and IR) is run first to elicit the agents’ private information, and then a classifier is applied to these values, which is presented as “allocating accuracy.” This leads to concerns of how much and how many agents must pay, which the authors answer with two theorems. The first shows that, when using SVMs as the classifiers, the expected number of payers is constant in the number of samples. The second is a negative result showing that k-nearest-neighbor classifiers can result worst case in the number of payees being linear in the number of samples. Since calculating payments in general are not tractable, algorithms are provided, both deterministic and randomized, to approximately calculate the correct prices. Finally, experiments are provided to enforce the theoretical results regarding the number of payees on synthetic data and the improvement in welfare on real data.

**Compliance With Llm Reviewing Policy:**

Affirmed.

**Final Justification:**

The authors generalized their theoretical results in the rebuttal, which greatly improves the significance of their work. They also added experimental results on runtime which are satisfactory. These updates together solidify the strength, both theoretically and in practice, of their algorithm. As such, I am happy to change my score to a 5.

**Key Questions For Authors:**

1. What classifiers are used in experiments?
2. The fact that only certain agents need to pay is unintuitive, could you provide some intuition for this? And is there a similar condition for which agents pay with the k-NN classifier?
3. Can you show that the improvements in welfare on the real data are statistically significant? Do you have confidence intervals? Can you also discuss the tradeoff of the welfare increase and the increase in computation cost incurred by running the auction; how does the auction cost compare to just classification?
4. Experiments are stated as using exact payment algorithm, but seems like binary search only gets epsilon approximation of the critical bid, can you clarify?

**Limitations:**

Yes

**Strengths And Weaknesses:**

Strengths:

- Studying an original and well motivated problem within strategic classification
- Using the standard assumptions of strategyproof and IR, I like this application of the classic Myerson’s lemma
- Theoretical results are elegant and use interesting methods for proof from other related settings (probabilistic optimal partitioning, stability), draw nice parallels with non-strategic classification
- Experiments on synthetic data use interesting techniques to demonstrate relationship between accuracy and paying users

Weaknesses:
- Framing the sum of bidder payments as revenue is misleading, isn’t the goal to minimize this?
- No distinction between types of misclassification (ex. False positives and false negatives can have very different effects in many settings like disease diagnoses)
- It is not clear what classifiers are being used in the experiments
- The improvement in welfare at the expense of some accuracy and the additional runtime from the auction is quite small (~2%)

Small notes:
- Notation b_j^{(i)} and b_j^c is confusing
- Typo in Figure 2 right-most: y=1 label for both blue and orange

---

> ### Author Rebuttal · Authors · 2026-03-31
>
> Thank you for your encouraging review!
>
> Weaknesses:
>
> > Framing the sum of bidder payments as revenue is misleading, isn’t the goal to minimize this?
>
> Thank you for pointing this out, and apologies for any confusion. By revenue we simply mean the sum of payments, not profit. The objective is to maximize welfare, towards which payments are an undesired yet necessary artifact. Ensuring that total payments are small, as in Thm. 2, is therefore indeed desirable. Note that optimizing welfare also means that payments need not be implemented as transfers, and can instead reduce utility indirectly, e.g. by injecting noise into predictions. To clarify, we will reframe the discussion to be in terms of social and residual surplus, rather than revenue.
>
> > No distinction between types of misclassification
>
> Our approach does support this wlog; please see our response to Rev. CVLk.
>
>
> > The improvement in welfare at the expense of some accuracy and the additional runtime from the auction is quite small (~2%)
>
> The gain in welfare that can be expected depends on the effect that values, as example weights, can have on learning outcomes. For the results shown in Sec. 7.2, in which values are set according to one of the dataset features, the effect is indeed mild (though statistically significant—see comment below). However, in Sec. 7.3 where values depend on labels, **welfare increases by over 20%**; see Fig. 4 in Appx. D3. Note that even in Sec. 7.2 gains increase with larger $v_+$, and reach up to 5% if we increase $v_+$ further. We will add these new values to Fig. 2, and move Fig. 4 to the main paper using the extra page.
>
>
> Questions:
>
> > What classifiers are used in experiments?
>
> Sec. 7.1 and Appendix D.2 use linear SVM. Sec. 7.2 and Appendix D.3 use logistic regression with $\ell_2$ regularization; please see our comment on generalizing Thm. 2 to apply to logistic regression in our response to Rev. phhN. Note Appendix C includes experimental details, to which we will make sure to add all of the above details.
>
> > The fact that only certain agents need to pay is unintuitive, could you provide some intuition?
>
> Lemma 2 states that points that pay can only be those that (i) obtain correct predictions, and (ii) are within distance $\beta_i$ of the decision boundary. Any point $i$ outside this region has no influence on allocations; economically, this means that they impose no (virtual) externalities on others, and so need not pay. This is conceptually similar to the idea of support vectors (which determine the learned classifier), but applied to the auction (which determines allocations and payments), and providing a sufficient (but not necessary) condition; indeed, in our experiments, even within this region the vast majority of points do not pay.
>
>
> > Is there a similar condition for which agents pay with the k-NN classifier?
>
> This is an excellent question. The conditions for paying with the k-NN classifier are in fact given by a closed form formula: An agent $(w,x,y)$ pays with the k-NN classifier if the aggregate label of its neighbors $\sum_{(w’,x’,y’)\in N_k(x)\setminus \{(w,x,y)\}} w’ * y’$ is smaller in magnitude and opposite in sign compared to $w y$ (where $N_k(x)$ is the set of neighbors, and $y\in\\{-1,1\\}$). We have added the proof to the paper.
>
> > [Are] the improvements in welfare on the real data statistically significant? Do you have confidence intervals?
>
> All results are statistically significant with p-values < 0.002. For all $v_+$, we ran a paired one-sided t-test comparing $\alpha=0$ and $\alpha=1$ over the $N=10$ random trials. For $v_+$ between 4 and 12, p-values ranged from 0.0018 to 0.00006, and 95% confidence interval sizes from 0.0053 to 0.0106, respectively.
>
>
> > Can you discuss the tradeoff of the welfare increase and the increase in computation cost incurred by running the auction?
>
> Computing payments requires, in the worst case, solving a series of classification sub-tasks. We provide two ways to reduce runtime. The first is to use classification-stable linear learning algorithms. These guarantee that: (i) the number of paying users is small and constant, (ii) testing candidate users is cheap, and (iii) sub-tasks require a much smaller sample size. Our experiments show that this reduction is significant and overall runtime is feasible: even for $m=10k$, logistic regression takes minutes, and SVM takes a few hours. We will add a table with all runtimes to the Appendix.
> The second is a choice between three pricing algorithms enabling different tradeoffs between compute and variance. In particular, Algo. 4 requires solving only two classification tasks in total.
>
>
> > Seems like binary search only gets epsilon apx. of the critical bid
>
> Since allocations are a step function, binary search can find an $\epsilon$-approximation for any $\epsilon$ (up to numerical precision) in time $O(\log(1/\epsilon))$. We used $\epsilon=10^{-10}$, and verified that smaller $\epsilon$ had no effect on outcomes.

---

> > ### Author Rebuttal · Reviewer_TSuQ · 2026-04-02
> >
> > Thank you for your thorough response to all of my concerns. I like the generalization to other values for error and payment for k-NN. I think these, along with the reframing of revenue, will go a long way in strengthening the paper. I will adjust my score accordingly.

---

> > > ### Author Response · Authors · 2026-04-07
> > >
> > > Thank you for the feedback! We are very grateful for the discussion, and will incorporate the new results and insights into the revised paper.

---

### Official Review · Reviewer_CVLk · 2026-03-12

**Soundness:** 4
**Presentation:** 4
**Significance:** 3
**Originality:** 4
**Overall Recommendation:** 5
**Confidence:** 5

**Summary:**

The paper studies classification with the objective of maximizing welfare instead of standard predictive accuracy. This is motivated by the fact that different users benefit by different values from being classified correctly. When these values are known, the welfare problem can be viewed as a weighted classification problem that prioritizes users with high values with a regularization term. However, these values are typically users' private information. This paper then introduces the so-called accuracy auction to incentivize users to truthfully report their values. The auction is as follows: users submit bids, and the classifier is trained using bids as weights, and payments are constructed (Myerson's lemma) to make truthful bidding incentive compatible. Based on the stability result of SVM and a probabilistic optimal partitioning argument, they prove that revenue (cost of information) is bounded for SVM, while for $k-$NN it grow linearly with sample size. The paper also gives exact and approximation payment computation methods and provided numerical experiments with synthetic and real data showing the tradeoff between welfare and accuracy.

**Compliance With Llm Reviewing Policy:**

Affirmed.

**Final Justification:**

I think it is a very novel and well-written paper, I recommend accepting it.

**Key Questions For Authors:**

1. I am wondering how the auction mechanism and revenue bounds might change if the utility function accounted for asymmetric harms, or is the single-parameter assumption a fundamental requirement for the application of Myerson's Lemma in this context?
2. The constant revenue upper bound holds for SVM. Do the authors have empirical evidence on the performance of other classifiers (e.g., logistic regression)?
3. The current framework is restricted to binary classification. Can the authors briefly comment on whether the framework is expected to extend naturally to multi-class settings?

**Limitations:**

Yes.

**Strengths And Weaknesses:**

Strengths

- The welfare-optimal classification is interesting. It pushes classification away from a purely statistical objective to a social allocation perspective. The paper does a good job of connecting machine learning with mechanism design by allowing users to truthfully report the weights used in learning.
- The constant revenue upper bound for SVM is surprising. The linear growth rate of $k$-NN is also appreciated, as it shows that classifier choice afftects not only prediction performance but also the cost of implementing accuracy auction.
- The exact and approximation methods of computing payments are provided.
- The question is overall very interesting and novel, and the approach appears to correct and intuitive. It also properly builds off of tools from the literature, which I appreciate.

Weaknesses

- User valuation is assumed to be $v \cdot \mathbf{1} \{ y = \hat{y}\}  $, meaning that all errors have 0 values. In many tasks, errors carry asymmetric negative costs, e.g., False Positives vs False Negatives.
- There is some missing explanation or connection discussion when moving from section 4 to the revenue discrepancies in section 5. On a first reading, I was a bit confused why do we care about revenue when maximizing welfare.
- It would be beneficial to explicitly list all the assumption, i.e., Assumption 1, etc.
- The paper assumes accuracy is a limited resource that requires allocation. While this is true under fixed data, it may be too restrictive in settings where better data  can improve accuracy for everyone simultaneously.

---

> ### Author Rebuttal · Authors · 2026-03-31
>
> Thank you for the encouraging and thorough review! We address your questions and comments below.
>
> > User valuations assume that all errors have 0 value
>
> Thank you for pointing this out! After investigating, we managed to show that **our framework extends naturally to auction settings with non-zero value of errors**, where agents are asked to report the difference between their valuations of accurate and inaccurate predictions.
>
> Formal intuition: For an agent corresponding to a feature-label pair $(x,y)$, denote by $v^{acc}$ their valuation of an accurate prediction, and by $v^{err} < v^{acc}$ their valuation of inaccurate prediction (note that $v^{err}$ may depend on $y$, capturing the difference between false positives/negatives). Equation (1) extends to $u(y,\hat{y}; v^{acc}, v^{err}) = (v^{acc} - v^{err}) \mathbb{I}[y=\hat{y}] + v^{err}$, which is pseudo-linear in the difference between the valuations, and shifted by a constant that is independent of the allocation. The constant shift in utility does not affect DSIC auction equilibria, as dominant-strategy equilibria are invariant to constant shifts in payoffs, and does not affect welfare maximization due to linearity. Therefore, asking the agents to report their benefit from receiving an accurate prediction ($v^{acc}-v^{err}$) induces a single-parameter auction setting identical to the one we analyze, and the setting we currently present in Section 3 coincides with $v=v^{acc}>0$,$v^{err}=0$. We will add this perspective to the paper, together with a formal analysis.
>
> > I was a bit confused why do we care about revenue when maximizing welfare
>
> Thank you for pointing this out, and apologies for the confusion. By revenue we simply mean the sum of payments, not profit. Indeed, the objective is to maximize welfare, and minimal payments are a desired property. To clarify, we will instead use “sum of payments” instead of revenue and reframe the discussion to be in terms of social and residual surplus.
>
> >“It would be beneficial to explicitly list all the assumption”
>
> Thanks for this suggestion, we will gladly add this to the final version.
>
> > The paper assumes accuracy is a limited resource that requires allocation. While this is true under fixed data, it may be too restrictive in settings where better data can improve accuracy for everyone simultaneously.
>
> This is an interesting idea. Indeed, if perfect accuracy is attainable, then there is no resource scarcity. If we understand your point correctly, then improving potential accuracy by improving data can be thought of as increasing the amount of (maximal) available resources, while our auction provides a method for efficiently allocating existing resources. Interestingly, this raises new questions regarding the optimal allocation of *effort* towards improving data. For example, assume data improvement is costly and the system is operating under a fixed budget; how should it allocate this budget across users in order to maximize welfare *after* the data been improved? We think this is a natural and intriguing follow-up direction.
>
> > The constant revenue upper bound holds for SVM. Do the authors have empirical evidence on the performance of other classifiers (e.g., logistic regression)?
>
> Theoretically, **we were able to generalize Thm. 2 and establish a constant payment upper bound for a much broader class of linear classification algorithms**, which includes logistic regression. Specifically, it holds for any convex Lipchitz loss with sufficient smoothness near the origin minimization, covering most common loss functions. For more details please see our response to Rev. phhN above.
>
> Empirically, we added new experiments measuring payments for increasing $m$ (up to 10k) on real data using the hinge loss, squared hinge, and log loss. Results supported our generalized theoretical bound, with the number of paying users rarely exceeding 20.
>
>
> > The current framework is restricted to binary classification. Can the authors briefly comment on whether the framework is expected to extend naturally to multi-class settings?
>
> From our initial investigation, it seems that extending our framework to the multi-class case is possible in some cases but not immediate in others.
>
> In more detail, the existence of DSIC accuracy auctions for binary classification relies on Theorem 1, which guarantees monotonicity of allocation for a large class of binary classification objectives. However, the proof of the theorem assumes that a point is classified correctly if and only if its loss is below a threshold - A property which holds for the multi-class 0-1 loss, but does not hold in general for smooth loss functions such as cross entropy. Thus, guaranteeing monotonicity in such settings seems to require additional assumptions about the dataset or algorithm, which are not required for binary classification. As such, multi-class accuracy auctions are an intriguing direction for future work, and we will highlight this in our discussion.

---

> > ### Author Rebuttal · Reviewer_CVLk · 2026-04-02
> >
> > Thank you for the rebuttal, I keep my score as is.

---

> > > ### Author Response · Authors · 2026-04-07
> > >
> > > Thank you again for the support and for the thoughtful questions! We will incorporate the new results into the revised paper.

---

### Official Review · Reviewer_phhN · 2026-03-19

**Soundness:** 3
**Presentation:** 3
**Significance:** 3
**Originality:** 3
**Overall Recommendation:** 5
**Confidence:** 3

**Summary:**

In this paper, the authors propose the welfare-optimal classification, a framework that prioritizes social welfare over predictive accuracy by modeling correct predictions as an auction in which a scarce resource is allocated based on users’ private valuations. Its main theoretical contribution is a novel link between algorithmic stability and auction theory, which shows that for linear SVM, the number of users who need to pay is $O(1)$ and does not grow with the training set size. In contrast, the paper proves an $\Omega(m)$ lower bound for k-NN, showing that this phenomenon is not universal and suggesting that algorithmic stability is the key factor governing welfare cost. The authors also introduce three payment algorithms that span different trade-offs between computational efficiency and strength of guarantees. Experiments on synthetic and semi-synthetic data, using one of these algorithms, provide empirical support for the proposed framework.

**Compliance With Llm Reviewing Policy:**

Affirmed.

**Final Justification:**

I am satisfied with the response and will change my score to 5.

**Key Questions For Authors:**

See the weakness part.

**Limitations:**

yes

**Strengths And Weaknesses:**

**Strengths:**

1. The paper introduces a clean and well-motivated framework that bridges mechanism design and machine learning.
2. The theoretical analysis is rigorous, with a clean proof strategy that repurposes classical notions of algorithmic stability to bound auction payments.
3. The paper presents a versatile set of payment algorithms spanning the trade-off between computational efficiency and strength of guarantees, with theoretical refinements that improve practical tractability.

**Weaknesses:**

1. The assertion in Lines 426-427 appears to overgeneralize by attributing the results to "stable learning algorithms" broadly, whereas the theoretical scope seems to be limited to classification stability, as stated in Section 5.1. For example, algorithms such as k-NN have at least $O(k/m)$ hypothesis stability [1,2], yet they do not satisfy the paper's cost-benefit conclusion. Without the qualifier "classification," the claim is not fully supported by either the literature or the paper's own analysis.
2. While the paper suggests that the result applies to a broad class of classification-stable algorithms, Theorem 2 is proved only for linear SVMs and does not provide a general proof framework that accounts for the diversity of algorithms within this class.
3. Theorem 2 relies on SVM-specific properties that do not appear to generalize directly. Could the authors provide, even in the form of a proof sketch, conditions under which the $O(1)$ payment bound extends to other classification-stable algorithms such as logistic regression, which is already used in the experiments but is not covered theoretically? More broadly, is $O(1/m)$ classification stability alone sufficient, or are additional algorithm-specific properties required?
4. For the randomized algorithm (Algorithm 3), DSIC holds only in expectation, which is a strictly weaker guarantee. In particular, a risk-averse user may prefer to misreport their valuation in order to reduce payment variance, even at the cost of lower expected utility. To justify the behavioral robustness of this randomized approach, could the authors clarify the assumptions on user utility functions? Specifically, does the mechanism remain incentive compatible under non-linear utility functions?

[1] Elisseeff, Andre, et al. "Stability of Randomized Learning Algorithms." *Journal of Machine Learning Research* 6.1 (2005).

[2] Bousquet, Olivier, and André Elisseeff. "Stability and generalization." *Journal of Machine Learning Research* 2.Mar (2002): 499-526.

---

> ### Author Rebuttal · Authors · 2026-03-31
>
> Thank you for your careful reading and detailed response. Please find our response to your comments below. In particular, **we have extended our main result in Thm. 2 to support a much larger class of learning algorithms**, which includes logistic regression; see details below. We hope this helps address what we believe is your main concern (weakness 2+3).
>
> > The assertion in Lines 426-427 appears to overgeneralize by attributing the results to "stable learning algorithms" broadly.
>
> We agree, and apologize if this unintentionally gave the wrong impression. We will make sure to state the claim in the discussion more precisely and in line with our results in Sec. 5.
>
>
> > Theorem 2 is proved only for linear SVM … and relies on SVM-specific properties. Could the authors provide conditions [that] extend to other classification-stable algorithms such as logistic regression?
>
> This is an excellent point! We managed to significantly improve this result, and extend our constant revenue upper bound to all classification-stable algorithms over linear classes with convex loss functions that are admissible (in the sense of Bousquet & Elisseeff 2002) and sufficiently smooth near the origin and with L2 regularization. This includes most common loss functions, including logistic regression (via binary cross entropy/log loss).
>
> The main generalization is of Lemma 5: We show that the signal strength (Definition 4) corresponds to the gradient of the aggregate loss function at the origin, and then use smoothness near the origin to derive a high-probability lower bound on the magnitude of the learned feature vector. Using this instead of the original Lemma 5 enables bounding the sum of payments for the larger class of learning algorithms described in a similar way as for SVM before.
>
> With this as motivation, we also added a new experiment that measures the number of paying users as a function of sample size that uses the folktables dataset, using the hinge loss (SVM), logistic regression, and the squared hinge. Under all algorithms, the number of paying users is at most $\sim 20$, even for $m=10000$, which is in line with Thm. 2. We will add these to the paper.
>
>
> > For the randomized algorithm (Algorithm 3), DSIC holds only in expectation, which is a strictly weaker guarantee… A risk-averse user may prefer to misreport their valuation in order to reduce payment variance. …  Could the authors clarify the assumptions on user utility functions? Specifically, does the mechanism remain incentive compatible under non-linear utility functions?
>
> The incentive compatibility guarantees of Algorithm 3 inherit from the original analysis of  Archer & Tardos (2004). Namely, DSIC holds when agents are risk-neutral (in the sense of having a linear “utility of money”), and make strategic decisions ex-ante (i.e. maximize expected pay before randomization). When agents are risk-averse (in the sense of having a concave utility of money) or require ex-post incentive compatibility (which is stricter than ex-ante), the payments computed by Algorithms 2 and 4 guarantee dominant strategy incentive compatibility (DSIC) regardless of agents’ risk attitude. We will clarify this point in the paper.

---

> > ### Author Rebuttal · Reviewer_phhN · 2026-04-02
> >
> > I thank the authors for their response. I will adjust my score accordingly.

---

> > > ### Author Response · Authors · 2026-04-07
> > >
> > > Thank you again for the encouraging feedback, and for the very helpful suggestions! We will gladly incorporate the new results and insights into the revised paper.

---

### Decision · Program_Chairs · 2026-04-30

**Decision:**

Accept (spotlight)

**Comment:**

The paper introduces accuracy auctions, a framework for welfare-optimal classification that elicits users' private valuations for correct predictions via a truthful auction mechanism. The key theoretical result is that for classification-stable algorithms, the expected number of paying users is bounded by a constant independent of sample size. Three reviewers scored the paper at 5, finding the contribution novel, rigorous, and at an interesting intersection of mechanism design and machine learning. During the rebuttal, the authors significantly strengthened the paper by generalizing the main theorem from SVM to a broad class of convex loss functions including logistic regression.

Reviewer FzyY's (who scored this paper a 3) primary concerns stem from practical and ethical considerations, which the authors acknowledge. However, as they also point out, this is a general criticism of many work in this space and not unique to their methods. I found their response to the reviewers' concerns adequate.

Overall, I think this will make a strong submission to the NeurIPS program.